# Barcode activity in a recurrent network model of the hippocampus enables efficient memory binding

Ching Fang[1†], Jack W Lindsey[1†‡], Larry F Abbott[1], Dmitriy Aronov[1,2], Selmaan N Chettih[1*]

[1]Zuckerman Mind Brain Behavior Institute, Columbia University, New York, United States; [2]Howard Hughes Medical Institute at Columbia University, Chevy Chase, United States

*For correspondence: selmaan.chettih@gmail.com

†These authors contributed equally to this work

Present address: ‡Anthropic, New York, United States

**Competing interest:** The authors declare that no competing interests exist.

## eLife Assessment

This **fundamental** work substantially advances our understanding of episodic memory by proposing a biologically plausible mechanism through which hippocampal barcode activity enables efficient memory binding and flexible recall. The evidence supporting the conclusions is **convincing**, with rigorously validated computational models and alignment with experimental findings. The work will be of broad interest to neuroscientists and computational modelers studying memory and hippocampal function.

**Abstract** Forming an episodic memory requires binding together disparate elements that co-occur in a single experience. One model of this process is that neurons representing different components of a memory bind to an 'index' — a subset of neurons unique to that memory. Evidence for this model has recently been found in chickadees, which use hippocampal memory to store and recall locations of cached food. Chickadee hippocampus produces sparse, high-dimensional patterns ('barcodes') that uniquely specify each caching event. Unexpectedly, the same neurons that participate in barcodes also exhibit conventional place tuning. It is unknown how barcode activity is generated, and what role it plays in memory formation and retrieval. It is also unclear how a memory index (e.g. barcodes) could function in the same neural population that represents memory content (e.g. place). Here, we design a biologically plausible model that generates barcodes and uses them to bind experiential content. Our model generates barcodes from place inputs through the chaotic dynamics of a recurrent neural network and uses Hebbian plasticity to store barcodes as attractor states. The model matches experimental observations that memory indices (barcodes) and content signals (place tuning) are randomly intermixed in the activity of single neurons. We demonstrate that barcodes reduce memory interference between correlated experiences. We also show that place tuning plays a complementary role to barcodes, enabling flexible, contextually appropriate memory retrieval. Finally, our model is compatible with previous models of the hippocampus as generating a predictive map. Distinct predictive and indexing functions of the network are achieved via an adjustment of global recurrent gain. Our results suggest how the hippocampus may use barcodes to resolve fundamental tensions between memory specificity (pattern separation) and flexible recall (pattern completion) in general memory systems.

## Introduction

Humans and other animals draw upon memories to shape their behaviors in the world. Memories of specific personal experiences — called episodic memories (*Tulving, 1972*) — are particularly important for livelihood. In animals, episodic-like memory is operationally defined as the binding

of the 'where', 'what', and 'when' components that comprise a single experience. This information can later be retrieved from memory to affect behavior flexibly, depending on context (*Clayton and Dickinson, 1998*). The binding of memory contents into a discrete memory is thought to occur in the hippocampus. Previous work proposed that the hippocampus supports memory by generating an 'index', that is, a signal distinct from the contents of a memory (*Teyler and DiScenna, 1986*; *Teyler and Rudy, 2007*). In this scheme, during memory formation, plasticity links the neurons that represent memory contents with the neurons that generate this index. Later, re-experience of partial memory contents may reactivate the index, and reactivation of the index drives complete recall of the memory contents.

Indexing theory was originally articulated at an abstract level, without reference to particular neural representations (*Teyler and DiScenna, 1986*). More recently, signatures of index signals were identified in neural activity through experiments in food-caching chickadees (*Poecile atricapillus*), an influential animal model of episodic memory (*Sherry, 1984*). *Chettih et al., 2024* identified 'barcode'-like activity in the chickadee hippocampus during memory formation and suggested that barcodes function as memory indices. Barcodes are sparse, high-dimensional patterns of hippocampal activity that occur transiently during caching. They are unique to each cache and are uncorrelated between cache sites, even for nearby sites with similar place tuning. Barcodes are then reactivated when a bird retrieves the cached item. Chickadee hippocampus also encodes the bird's location — as expected, given the presence of place cells — as well as the presence of a cached sunflower seed, irrespective of location. Thus, *Chettih et al., 2024* found that hippocampal activity contains both putative memory indices (in the form of barcodes) and putative memory content (in the form of place and seed-related activity).

These findings raise several critical questions. How are barcodes generated and associated with place and seed-related activity during caching? How can hippocampal dynamics subsequently recall these same patterns during retrieval? Critically, neurons participate in both barcodes and place codes with near-random overlap, in contrast with theoretical models where content and indexing functions occur in separate neurons (*Krotov and Hopfield, 2016*; *Bricken and Pehlevan, 2021*; *Tyulmankov et al., 2021*; *Whittington et al., 2021*; *Kozachkov et al., 2023*). It is unclear at a computational level how memory index and content signals can be functionally distinct when they coexist in the same network and even in the activities of single neurons.

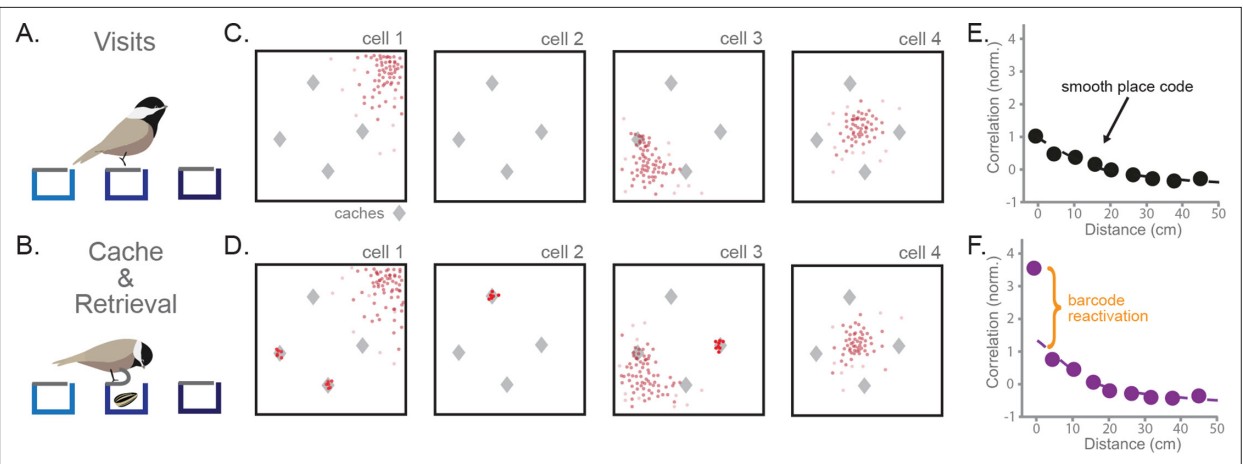

**Figure 1.** Hippocampal activity during food-caching reveals sparse 'barcodes' for each cache memory. (**A**) A black-capped chickadee visits cache sites in a laboratory arena. The chickadee does not interact with cache sites during what we call a visit. (**B**) A chickadee caches or retrieves at a cache site by peeling back the rubber flap at the site. This reveals a hidden compartment where seeds can be stored. (**C**) Cartoon example of spiking activity during visits of four hippocampal neurons in a square arena. Spikes are in red. Gray diamonds indicate the location of sites with caches. (**D**) As in (**C**), for the same cells during caching and retrieval. Neurons fire sparsely and randomly during caches (activity clusters at sites) independent of their place tuning. (**E**) Correlation of population activity during visits to the same or different sites of increasing distance. Values are max-normalized. Reproduced with permission from *Chettih et al., 2024*. (**F**) As in (**E**), but comparing activity from caches at a site to activity from retrievals at the same or different sites. Barcode activity shared between caches and retrievals at the same site produces a sharp deviation from smooth spatial tuning. Values are normalized by the same factor as in (**E**). Reproduced with permission from *Chettih et al., 2024*.

In this paper, we use the findings of *Chettih et al., 2024* to guide the design of a biologically plausible recurrent neural network (RNN) for cache memory formation and recall. The model generates barcodes and associates them with memory content in the same neural population. At recall time, the model can flexibly adjust the spatial scale of its memory retrieval, ranging from site-specific information to search of an extended area, depending on contextual demands. Using this model, we demonstrate the computational advantages of barcode-mediated memory by showing that it reduces interference between similar memories. We also show that place and barcode activities in the model play complementary roles in enabling memory specificity and flexible recall.

## Results

### Barcode representations in the hippocampus are observed during caching

We first review the key experimental results in *Chettih et al., 2024*. Black-capped chickadees were placed in a laboratory arena comprised of a grid of identical sites. Each site had a perch to land on and a hidden compartment, covered by a flap, where a seed could be hidden. Chickadees were allowed to behave freely in this environment and collect sunflower seeds from feeders. A chickadee often visited sites without interacting with the hidden compartment (*Figure 1A*) but, at other times, the chickadee cached a seed into the compartment, or retrieved a previously cached seed (*Figure 1B*). Previous experiments demonstrated that chickadees remember the precise sites of their caches in this behavioral paradigm (*Applegate and Aronov, 2022*).

*Chettih et al., 2024* recorded hippocampal population activity during these behaviors. When chickadees visited sites, place cells were observed, similar to those previously found in birds and mammals (*Payne et al., 2021*; *Figure 1C*). Place cells did not change their spatial tuning after a cache. Instead, during caching and retrieval, neurons transiently displayed memory-related activity. During caching, neurons fired sparsely, with individual neurons producing large activity bursts for a small subset of caches at seemingly random locations (*Figure 1D*). These bursts occurred in both place cells and non-place cells, with the location of cache bursts unrelated to a neuron's place tuning. At the population level, activity during a cache consisted of both typical place activity and a cache-specific component orthogonal to the place code, termed a "barcode". Strikingly, barcode activity for a particular cache reactivated during later retrieval of that cache. These findings were evident when examining the correlation between population activity vectors for visits, caches, and retrievals at different sites. When comparing two visits, the correlation profile decayed smoothly with distance, as expected from place tuning (*Figure 1E*). When comparing caching with retrieval, a similar smooth decay was observed for most distances, indicating the presence of place tuning. However, there was a substantial boost in correlation for caches and retrievals at the exact same site (*Figure 1F*). This site-specific boost resulted from reactivation of the cache barcode during subsequent retrieval.

Barcode activity during caching and retrieval, which are moments of memory formation and recall, suggests a mechanism supporting episodic memory. We hypothesize that the hippocampus generates a sparse, high-dimensional pattern of activity transiently during the formation of each memory, and that this serves as a unique index to which the contents of the memory are bound. Reactivation of the barcode at a later time drives the recall of the associated memory contents. In the food caching behavior studied by *Chettih et al., 2024*, the contents of memory include the location ('where') and presence of a seed ('what'). Below, we implement this hypothesis in a computational model.

### Generating barcode activity with random recurrent dynamics

We model the hippocampus as a recurrent neural network (RNN; *Alvarez and Squire, 1994*; *Tsodyks, 1999*; *Hopfield, 1982*) and propose that recurrent dynamics can generate barcodes from place inputs (*Figure 2A*). As in experiments, the model's population activity during a cache should exhibit both place and barcode activity components. The network receives place inputs termed $\vec{p}$ which are spatially correlated as follows. Each neuron in the RNN receives inputs that are maximal at one location in the environment and decay with distance from that location, causing the neuron to have a single place field centered at that location. Across neurons, place fields uniformly span the environment. The firing rate activity of RNN units is denoted as $\vec{x}$. The recurrent weights of the model are given by $J$, and the RNN activity $\vec{x}$ follows standard dynamics equations:

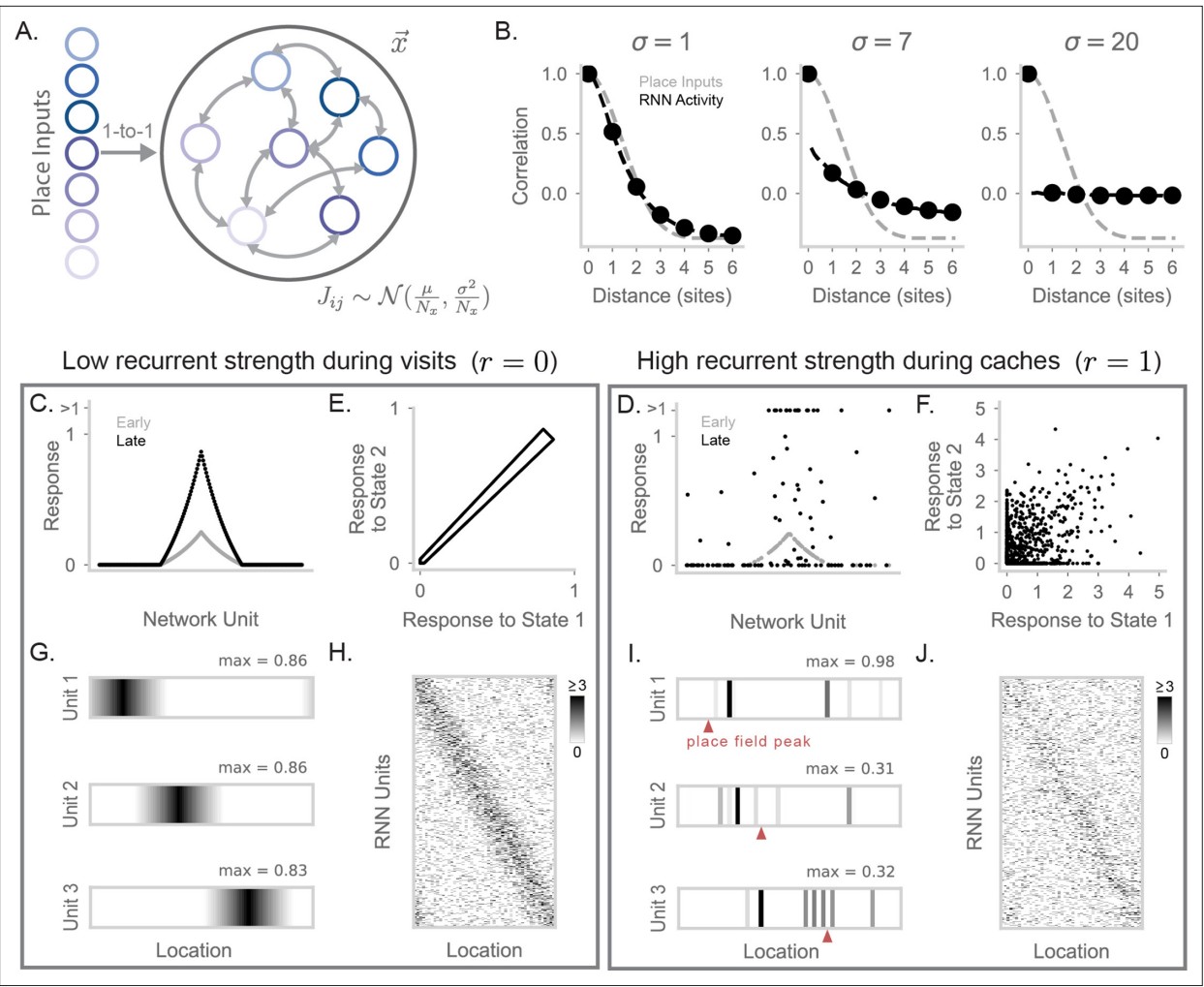

**Figure 2.** A recurrent neural network generates barcode activity through recurrent dynamics. (**A**) Diagram of a RNN; the activity of the network units is denoted as $\vec{x}$. Place information arrives from an input layer with activities $\vec{p}$. Recurrent weights are initialized randomly, that is, $J_{ij} \sim \mathcal{N}(\frac{\mu}{N_x}, \frac{\sigma^2}{N_x})$ for the synapse connecting neuron $j$ to neuron $i$, where $N_x$ is the number of RNN neurons. (**B**) Correlation of activity vectors across different locations, when RNN weights are initialized with $\sigma = 1$ (left), $\sigma = 7$ (center), and $\sigma = 20$ (right). We show correlation of place inputs (gray) and correlation of the RNN's rate vector at $t = 100$ (black). X-axis is in units of site distance (see Methods for definition). (**C**) Response of RNN units when simulating a visit to a location halfway around the circular track (with $r = 0$; **equation 3**). In gray is the activity of the RNN at $t = 1$. In black is the activity at $t = 100$ RNN units that are uniformly subsampled and sorted by the tuning of their inputs for plotting purposes. (**D**) As in (**C**), but for $r = 1$. RNN activity is more sparsely distributed, including high activity in neurons without place tuning for the current location. For visualization purposes, 50 RNN units with nonzero activity and 50 RNN units with 0 activity are sampled at this time point ($t = 100$) for display, and responses greater than 1 are clipped to $> 1$. (**E**) RNN activity in response to state 1 compared to state 2, when $r = 0$. Each point corresponds to a single RNN unit. (**F**) As in (**E**), but for $r = 1$. RNN activity for these neighboring states is substantially decorrelated by recurrence. (**G**) Firing fields of three example units on the circular track displaying place-cell activity. Maximum value of each field is labeled and the colormap is max-normalized. (**H**) Simulated spike counts of a RNN population during visits to each location on the circular track. Spikes are generated by simulating Poisson-distributed counts from underlying unit rates. Place tuning results in a strong diagonal structure when units are sorted by their input's preferred location. The maximum limit of the colormap is set to the 99th percentile value of spike counts (≥3 spikes). (**I**) Same neurons as (**G**), but for $r = 1$ with units now showing barcode activity. The location of the $r = 0$ place field peak of each unit (i.e. its corresponding peak in (**G**)) is marked by a red triangle. (**J**) As in (**H**), but for $r = 1$. The independence of barcode activity for neighboring sites results in a matrix with reduced diagonal structure.

The online version of this article includes the following figure supplement(s) for figure 2:

**Figure supplement 1.** Convergence of RNN dynamics and additional unit activity examples.

**Figure supplement 2.** Feedforward model comparison for barcode generation.

$$\vec{x} = ReLU(\vec{v}) \tag{1}$$

$$\frac{d\vec{v}}{dt} = -g(\vec{x})\vec{v} + J\vec{x} + \vec{p} \tag{2}$$

where $\vec{v}$ is the voltage signal of the RNN units and $g(\cdot)$ is a leak rate that depends on the average activity of the full network, representing a form of global shunting inhibition that normalizes network activity to prevent runaway excitation from recurrent dynamics. In our simulations, we simplify the task by using a 1D circular environment binned into 100 discrete spatial 'states'. We set the spatial correlation length scale of place inputs such that the smallest distance between cache sites in the experiments of *Chettih et al., 2024* is equal to 8 of these states (see Methods for more details).

We initialize the recurrent weights from a random Gaussian distribution $J \sim \mathcal{N}(\frac{\mu}{N_x}, \frac{\sigma^2}{N_x})$, where $N_x$ is the number of RNN neurons and $\mu < 0$, reflecting global subtractive inhibition that encourages sparse network activity to match experimental findings *Chettih et al., 2024*. We first consider network activity before any learning-related changes. For the range of parameters we use, the network is in a chaotic state with a roughly constant overall level of activity but fluctuations in the activities of individual units. From an initial state of 0, we run recurrent dynamics until this steady-state level of overall activity has been achieved (*Figure 2—figure supplement 1A, B*). The chaotic recurrent dynamics induced by the random weights (*Sompolinsky et al., 1988*) effectively scrambles incoming place inputs. We demonstrate this by measuring the correlation of RNN activity across different locations and plotting this correlation as a function of distance (*Figure 2B*). Place inputs show a smoothly decaying correlation curve. At low values of $\sigma$, the network is primarily input-driven, showing a smoothly decaying correlation matching inputs. At high values of $\sigma$, recurrence is so strong that it entirely eliminates the spatial correlation of nearby sites: activity at each state is decorrelated from activity at all other states.

Interestingly, at an intermediate level of $\sigma = 7$, the network retains elements from both high and low recurrence regimes (*Figure 2B*). The network exhibits a smoothly decaying correlation curve reflecting its inputs, but each state's activity also contains a strong decorrelated component, apparent in the large drop in correlations for nearby but non-identical sites. This intermediate network regime closely resembles spatial correlation profiles observed during caching (*Chettih et al., 2024*; *Figure 1F*). The smoothly decaying component is caused by the place code, whereas the sharp peak at zero distance – reflecting barcode activity – is caused by the recurrent dynamics. We thus construct networks using intermediate $\sigma$ values, which allow for the coexistence of place code and barcode components in population activity.

A key result from *Chettih et al., 2024* is that the hippocampus exhibits both place code and barcode activity during caching and retrieval, but only place activity during visits. We propose that this effect can result from a network in which recurrent strength is flexibly modulated. In a low-recurrence condition, the network produces the place code, whereas in a high-recurrence condition, the same network produces a combination of place code and barcode activity. To simulate this in our model, we modify *equation 2* to

$$\frac{d\vec{v}}{dt} = -g(\vec{x})\vec{v} + rJ\vec{x} + \vec{p} \tag{3}$$

where the newly included $r \in \{0, 1\}$ scales recurrent strength such that the network may be in a feedforward ($r = 0$) or recurrent regime ($r = 1$). During visits, we assume the network is operating with low recurrent strength ($r = 0$). As a result, the activity in the RNN exhibits the spatial profile of its place inputs. We verify this is the case by visualizing early and late RNN activity given a place input (*Figure 2C*). With low recurrence, early and late RNN activity have similar spatial profiles, and late activity patterns for nearby states are highly correlated (*Figure 2E*). In contrast, when recurrence is enabled ($r = 1$), network activity is sparse with a heavy tail of a few strongly active neurons and is decorrelated between nearby states (*Figure 2D and F*). These changes match experimental observations of excitatory neurons during food caching (*Chettih et al., 2024*).

We visualized neural activity during visits ($r = 0$) and caches ($r = 1$). During visits, RNN units display ordinary place tuning (*Figure 2G*, *Figure 2—figure supplement 1C*). We also simulated spikes for all units at each location and visualized this as a matrix where units were sorted by the preferred location of their inputs (*Figure 2H*). This matrix exhibits a diagonal structure reflecting strong spatial correlations. During caches, single neurons develop sparse, scrambled spatial responses relative to their place tuning (*Figure 2I*, *Figure 2—figure supplement 1C*). Accordingly, simulated

spikes across the population have greater random, off-diagonal components during caching than during visits. Thus, recurrence in a randomly connected RNN is a simple and effective mechanism to generate barcode signals from spatially correlated inputs. We alternatively considered a feedforward mechanism to generate barcodes, in which barcodes are computed by a separate feedforward pathway (*Figure 2—figure supplement 2*). We found the feedforward mechanism required an unreasonably large number of neurons and sparsity levels to match the decorrelation level of the recurrent mechanism.

## Storing memories by binding content inputs with a barcode

Having suggested a mechanism for the generation of barcode representations, we next propose how such a representation can be leveraged for memory storage in a network model. In our food-caching task, we assume that the contents of a memory include the location of the cache and the identity of a food item within the cache. Thus, we add an additional input source besides spatial location into our model – an input $s$ representing the presence of a seed (*Figure 3A*). The 'seed' input projects randomly onto the RNN neurons. During caching, both place inputs and seed inputs arrive into the RNN, matching experimental findings (*Chettih et al., 2024*). This causes a mixed response in the network: one component of the response (place activity) is spatially tuned, another component (seed activity) indicates the presence of a seed and does not vary with location, and the third component (barcode activity) is generated by recurrent dynamics interacting with these inputs.

To store memories, we assume the RNN undergoes Hebbian learning after some fixed time point during the $r = 1$ recurrent dynamics (*Figure 3A*). At this time, the synapse $j \rightarrow i$ changes by an amount

$$\Delta J_{ij} \propto x_i x_j - \beta x_i \qquad (4)$$

where $\beta > 0$ provides an overall inhibition of the stored pattern. This term helps to prevent network activity from collapsing to previously stored attractors. Memory storage works as follows, following the experimentally observed sequence in *Chettih et al., 2024*: place inputs arrive into the RNN, recurrent dynamics generate a random barcode, seed inputs are activated, and then Hebbian learning binds a particular pattern of barcode activity to place- and seed-related activity. In this way, fixed-point attractors are formed corresponding to a conjunction of a barcode and memory content.

Memory recall in our network follows typical pattern completion dynamics, with recurrence strength set to $r = 1$ as for caches. As an example, consider a scenario in which an animal has already formed a memory at some location $l$, resulting in the storage of an attractor $\vec{a}$ into the RNN. The attractor $\vec{a}$ can be thought of as a linear combination of place input-driven activity $p(l)$, seed input-driven activity $s$, and a recurrent-driven barcode component $b$. Later, the animal returns to the same location and attempts recall (i.e. sets $r = 1$, *Figure 3B*). Place inputs for location $l$ drive RNN activity towards $p(l)$, which is partially correlated with attractor $\vec{a}$, and the recurrent dynamics cause network activity to converge onto attractor $\vec{a}$. In this way, barcode activity $b$ is reactivated as part of attractor $\vec{a}$, along with the place and seed components stored in the attractor state $p(l)$ and $s$. The seed input can also affect recall, as discussed in the following section.

As an initial test of memory function in our model, we analyzed the activity patterns that are stored and recalled in the model. We simulated caching at different sites in the arena and extracted the population activity that is stored via Hebbian learning (*Figure 3C*, left). We then simulated retrieval as an event in which the animal returns to the same site and runs memory recall dynamics ($r = 1$) in the RNN. Population activity during retrieval closely matches activity during caching and is substantially decorrelated from activity during visits (*Figure 3C*). To compare our model with the empirical results reproduced in 1E,F, we ran in silico experiments with caches and retrievals at varying sites in the circular arena. We simulated Poisson-generated spikes drawn from our network's underlying rates to match the intrinsic variability in empirical data (see Methods). We find that population activity in the RNN is more strongly correlated between caches and retrievals at the same site than two visits to a site that are not seed related ($r = 0$; *Figure 3D*). In addition, cache-retrieval correlations for non-identical sites rapidly drop to the level of visit-visit correlations because barcodes are orthogonal even for nearby sites. These correlation profiles closely resemble those observed in *Chettih et al., 2024* (compare *Figure 3D* and *Figure 1E, F*).

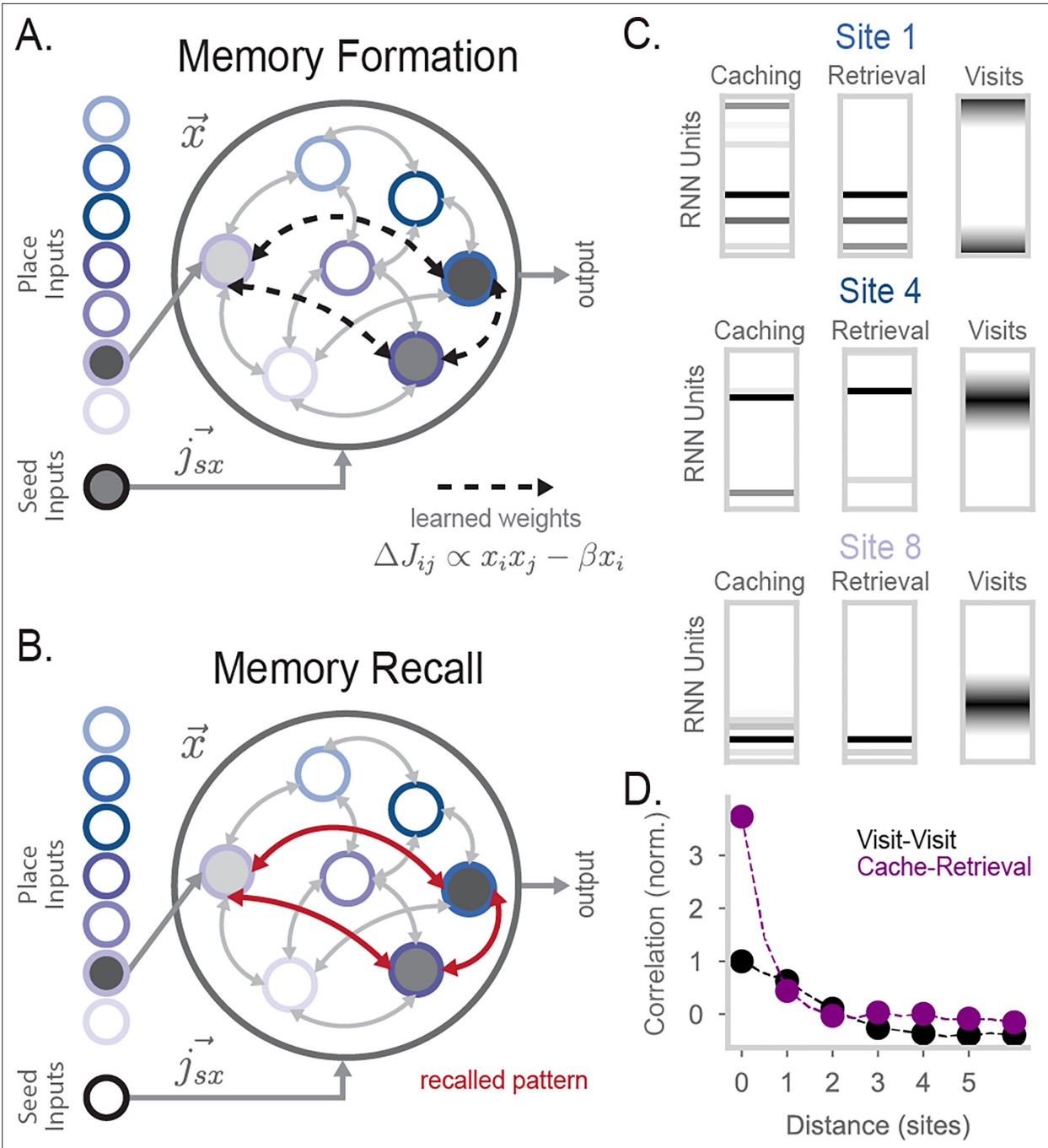

**Figure 3.** Barcode activity binds content to store memories in the RNN. (**A**) Diagram of RNN activity during memory formation. Along with place inputs, a scalar seed input $s$ is provided to the model. $s$ connects to RNN units via 5000-dimensional weight vector $\vec{j}_{sx} \sim \mathcal{N}(0, 1)$. During memory formation (i.e. when an animal caches a seed), place inputs representing the animal's current location and a seed input are provided to the RNN. After recurrent dynamics are run for some time ($t = 100$), the network undergoes Hebbian learning. (**B**) An example of memory recall in the network. The animal is at the same location as in (**A**). The place input encoding that location is provided to the model and results in the RNN recalling the stored attractor pattern seen in (**A**). C. Examples of RNN population activity during caching (left), retrieval (center), and visits (right) at three sites. During visits, the RNN has $r = 0$, while during caches and retrievals $r = 1$. For visualization purposes, 50 units are randomly sampled and displayed in the 'Caching' and the 'Retrieval' plot. (D) Correlation of Poisson-generated spikes simulated from RNN rate vectors at two sites, plotted as a function of the distance between the two sites. In black is the correlation in activity between two visits. In purple is the correlation between caching and retrieval activity. Experiments were simulated with 20 simulations where 5 sites were randomly chosen for caching and retrieval. 99% confidence intervals are calculated over the simulations and plotted over the markers. Compare to *Figure 1E and F*.

## Barcode-mediated memory supports precise and flexible context-based memory recall

What are the computational benefits of barcode-mediated memory storage? We designed two behavioral tasks for our model that quantify complementary aspects of memory performance. In both tasks, we simulate a sequence of three caches in the circular arena. We then test the model's performance during memory recall (i.e. $r = 1$) using two distinct tasks. The 'Cache Presence' task requires the network to identify the presence or absence of a cached seed at the current location (*Figure 4A*). By evaluating the model on this task using different spacings between caches, we can measure the spatial precision of memory, that is how far apart two caches must be to prevent interference between their associated memories. The 'Cache Location' task requires the network to identify the cache nearest to the current location by reactivating place activity corresponding to that cache location (*Figure 4B*). This task measures the robustness of recall and requires attractor dynamics to accurately retrieve memories from potentially distant locations. Together, these questions probe the ability of the memory system to be both specific (pattern separation) and searchable (pattern completion).

To enable readout from the model, we add an output layer containing place and seed units. During memory formation, these units receive a copy of place and seed inputs, respectively, and undergo Hebbian plasticity (*equation 4* with $\beta = 0$) with RNN units. This enables the RNN to reactivate a copy of the inputs that were provided during memory formation. We measure the activity of the seed output to determine cache presence, and we measure place outputs to determine cache location. We note that other readout mechanisms would likely function similarly – for example, plastic feedback connectivity from recurrent onto input units. The output layer used here is not intended to correspond to a specific brain region, but simply to provide a window into what could be read out easily from network activity by downstream neural circuits.

We first show model performance on a single example of the Cache Presence and Cache Location tasks, where caches are made at the locations 20%, 35%, and 70% of the way around the circular track. For the Cache Presence task, we evaluate the model at each spatial location and plot the activity of the seed output (*Figure 4C*). After the first cache, we see that the seed output is only high at states around the vicinity of the first cache (*Figure 4C*, top). The false positive rate is not zero since the states around the cache also have high output. However, the width of this false positive area is less than the distance between two sites in the arena used in *Chettih et al., 2024*, indicating that the model is able to identify caches at the spatial resolution observed in chickadee behavior in these experiments. After a second cache, the seed output correctly reports seeds in the areas around the two caches (*Figure 4C*, middle). Importantly, the network separates the caches from each other, correctly identifying the absence of a seed at a position between the two caches. This behavior is maintained after the addition of the final third cache (*Figure 4C*, bottom).

For the Cache Location task, we examine the place outputs (*Figure 4D*, 'Narrow Search'). When the current position is near a cache, the network correctly outputs a place field for the location of the nearest cache. However, an animal relying on memory to guide behavior may need to recall cache locations even when far away from a cache. To enable increased search radius, we make use of the seed input, which can bias network activity towards all memory patterns previously associated with this input. Activating the seed input greatly increases the range of current positions over which cache memories are retrieved (*Figure 4D*, 'Broad Search'). Critically, this does not cause interference between memories. For example, as the current position moves from the location of cache 1 to cache 2, place outputs discretely jump from one cache location to the other, corresponding to correct selection of the nearest cache location. The search radius during recall can thus be flexibly adjusted according to task demands, allowing the trade-off between pattern completion and pattern separation to be dynamically regulated by simple scaling of a network input.

We next systematically quantified model performance on these tasks. For these analyses, we binarized the seed output as being above or below a threshold (*Figure 4—figure supplement 1A, B*). When caches are sufficiently spaced, memory recall is near perfect (*Figure 4—figure supplement 1A, B*). This led us to evaluate the resolution of the model's memory as caches become more closely spaced. We decreased the distance between cache 1 and 2 to identify the minimum distance where caches are separable, that is where the model correctly indicates the absence of a cache at a midpoint between caches 1 and 2. *Figure 4E* shows this correct reject rate as a function of the distance from each cache and the seed input strength. With low seed input (e.g. $s = 0$), the correct reject rate is high

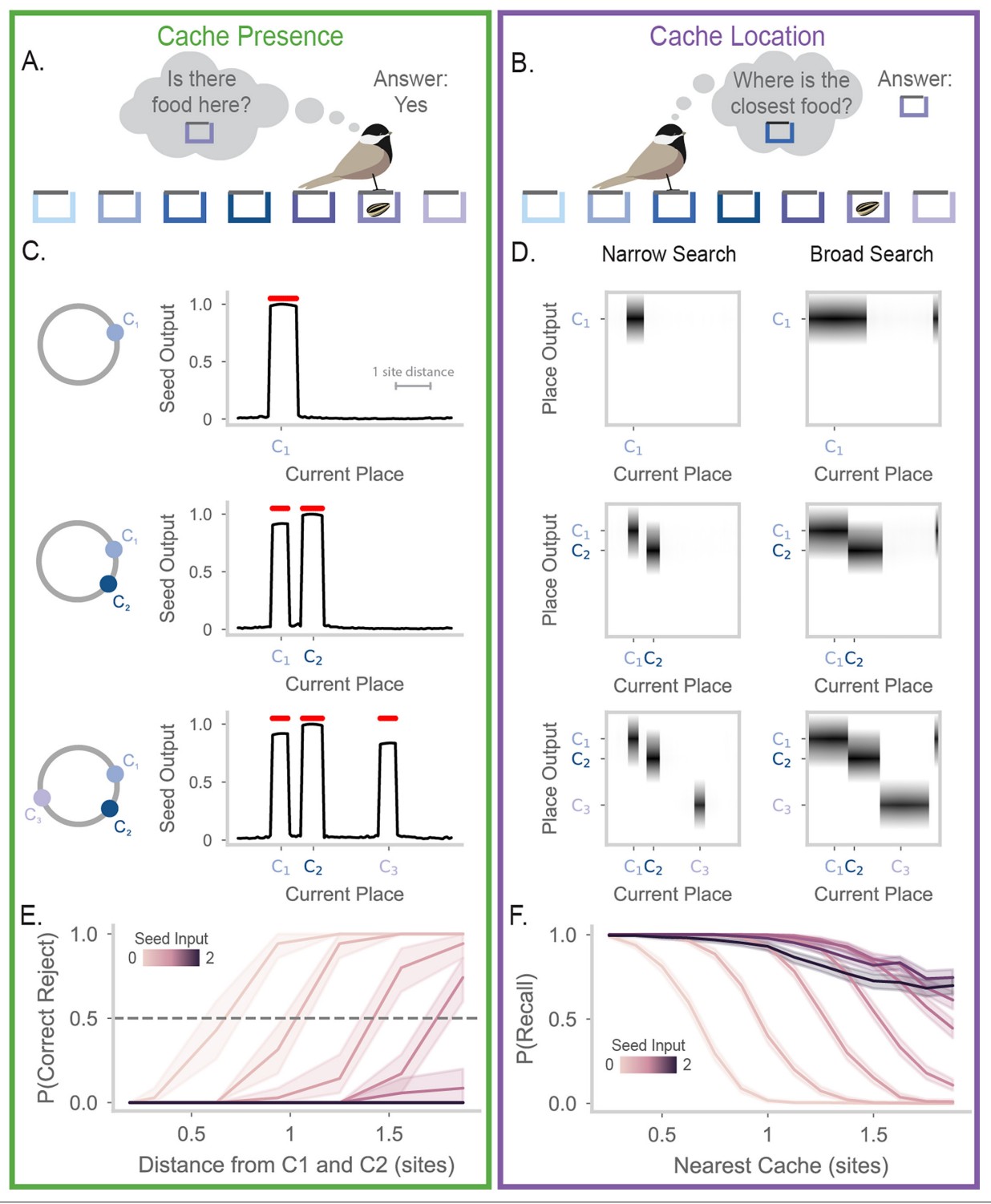

**Figure 4.** Precise and flexible recall of barcode-mediated cache memories. (**A**) Cartoon of a chickadee at a site, trying to remember whether the site contains a seed. The chickadee cannot see inside the hidden compartment and must rely on memory to answer this question. (**B**) Cartoon of a chickadee at a site, trying to recall the location of the closest cache. In this case, the animal must use its memory to recall the location of a cache three sites away. (**C**) Seed output of the RNN at different locations along the circular track. Red dots above the line indicate locations where the value is greater than 0.5. Top, results after the first cache is made at a location 20% of the way through the track. The location of the first cache is marked on the x-axis as $C_1$. Middle, same but after the second cache ($C_2$) is made at a location 35% of the way through the track. Bottom, same but after the third cache ($C_3$) is made at a location 70% of the way through the track. (**D**) The place output of the RNN at different locations along the track. In the

*Figure 4 continued on next page*

*Figure 4 continued*

heatmaps, each column shows the activities of all the output units when the animal is at a particular location (horizontal axis). The left side panels of the subfigure correspond to the model operating at low recall strength ($s = 0$). The right side panels correspond to the model operating at high recall strength ($s = 1.5$). As in (C), top, middle, and bottom plots correspond to RNN activity after caches are made at C1, C2, and C3. Place output correctly switches between the location of each cache depending on proximity to the current position. (E) Probability of correct reject rate in the model when the animal is at a location between caches 1 and 2, after all caches have been made. The X-axis shows the distance from the probed location to the surrounding caches, measured in site distance. The color of each line corresponds to the recall strength $s$. At low recall strengths, the model correctly identified the absence of a cache between C1 and C2, reflecting discrimination of the two cache locations. Experiments were simulated with 35 random seeds. 99% confidence intervals are calculated over the simulations and plotted as line shading. (F) Probability of recalling the location of the closest cache, given the distance of the animal from the cache. A recall is considered successful if the seed output exceeds 0.5 and if the peak of the output corresponds to the location of the nearest cache. Lines are colored by recall strength $s$. The model correctly recalls the locations of nearby caches, with search radius increasing as recall strength increases. Number of simulations and error shading as in (E).

The online version of this article includes the following figure supplement(s) for figure 4:

**Figure supplement 1.** Threshold selection and parameter sensitivity for seed output detection.

**Figure supplement 2.** Robustness of recall to noise in dynamics and place inputs.

**Figure supplement 3.** Model performance with realistic Gaussian place field inputs.

---

when the current location is one site distance away from caches 1 and 2. In other words, if a cache is made at site 1 and site 3, our model recognizes that site 2 remains empty. This precision matches the single-site resolution measured by behavioral experiments in this arena (*Applegate and Aronov, 2022*). As expected, performance on the Cache Presence task decreases with greater seed input strengths, as these are suited to searching over a broad spatial range.

For the Cache Location task, we measure the probability that the model outputs the location of the nearest cache. We plot this probability as a function of distance from the current location to the nearest cache (*Figure 4F*). The model is able to correctly recall cache locations with almost perfect accuracy when it is near a cache. Performance drops sharply with distance when the seed input is low, but is substantially recovered by increasing the seed input strength. Critically, even when the search radius is broad enough to include multiple caches, attractor dynamics encourage selection of the single closest cache location rather than blending memories. Thus, the seed input strength provides a flexible search radius during the recall process. Low values of $s$ are more suitable for detecting the presence or absence of a seed near the current location, while high values of $s$ are more suitable for finding remote caches. The complementary demands of the Cache Presence and Cache Location tasks demonstrate the utility of a flexible search radius within one memory system.

We further examined how model hyperparameters affected performance on these tasks. We find that the plasticity bias $\beta$ is needed to prevent erroneous memory recall at sites without caches (*Figure 4—figure supplement 1E-H*). Without this, recall specificity is poor and model performance suffers on the Cache Presence task. We also verified that our model is robust to the order in which caches were made (*Figure 4—figure supplement 1C, D*). We found that adding noise to the network's temporal dynamics had little effect on memory recall performance (*Figure 4—figure supplement 2A*). However, large static noise vectors added to the network's input and initial state decreased the overall probability of memory recall, but not its spatial profile (*Figure 4—figure supplement 2B*). Finally, we extended our model to work with random, spatially correlated inputs, rather than receiving place cell-like inputs (*Figure 4—figure supplement 3*). This shows that our results apply regardless of the specific format of spatial inputs, which are experimentally undetermined.

## Place activity and barcode activity play complementary roles in memory

We have shown that a barcode-mediated memory system is precise yet allows flexible, content-based retrieval. Below we identify the specific contributions of place and of barcode activity by ablating either of these components in our model. To ablate barcodes ('Place Code Only' in *Figure 5*), we initialize our model without the random recurrent weights that produce chaotic dynamics. This is akin to running caching dynamics in 'visit mode', that is $r = 0$ in *Figure 2*. To ablate place activity ('Barcode Only' in *Figure 5*), we eliminate spatial correlations in the network inputs. This is akin to having place fields with extremely narrow precision, rather than a spatially smooth place code. We test both ablated models on the three-cache tasks from above and compare their performance to our full model.

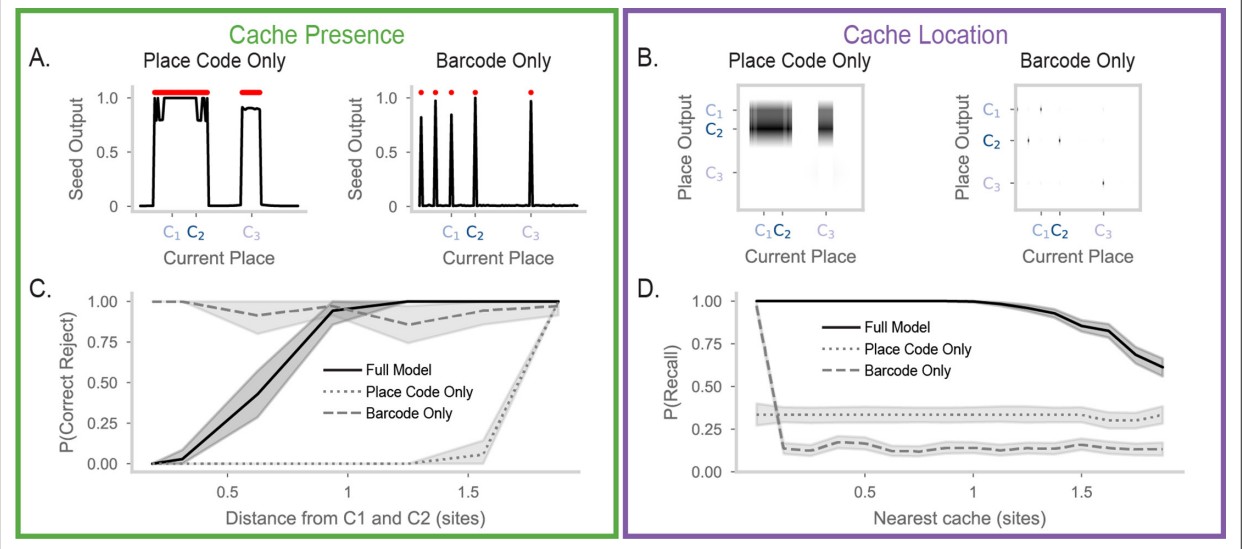

**Figure 5.** Ablations reveal complementary roles of place code and barcode activity in memory recall. (**A**) Left, as in *Figure 4C*, bottom, but for the 'Place Code Only' model (barcode-ablated), which has recurrent weights initialized with $\mu = 0, \sigma = 0$. Right, same, but for the 'Barcode Only' model (place-ablated). The barcode-only model is the same as the full model, but place inputs are uncorrelated for nearby locations. (**B**) Left, as in *Figure 4D*, bottom, but for the place code only model. Right, the same but for the barcode-only model (for visual clarity, the matrix is downsampled to remove zero-value rows). The place code-only model has similar outputs for recall from all current locations. The barcode only model has no place output for recall from most locations. (**C**) As in *Figure 4E*, but showing the full model (solid line), the place code only model (dotted line), and the barcode only model (dashed line). For all models, $s = 0$. The place code only model is unable to discriminate between nearby caches. (**D**) As in *Figure 4F*, with lines as in (**C**). For all models, $s = 0.4$. The barcode-only model recalls successfully only when the current location contains a cache. Only the full model reliably recalls remote cache locations.

The online version of this article includes the following figure supplement(s) for figure 5:

**Figure supplement 1.** Performance comparison of place-code-only and barcode-only models.

The Place Code Only model directly binds memory contents of place and seed. This causes cache memories for nearby locations, where the place code is correlated, to interfere with each other. Interference is clearly visible in the performance on the Cache Presence task (*Figure 5A*, left). The seed output of the place code only network is high across a wide area including caches 1 and 2 and the empty locations between them. Indeed, this network is unable to identify the absence of a cache at locations between caches 1 and 2, even with a substantial distance between them (*Figure 5C*). Furthermore, the memories for caches 1 and 2 sometimes appear to suppress memory for a distant cache 3. Further evidence of interference is apparent in the Cache Location task, where the network merges caches 1 and 2 and entirely fails to signal the cache at location 3 (*Figure 5B*, left). This network has a low true negative rate, and often a single cache dominates the output (*Figure 5—figure supplement 1A, D*). Accordingly, the network is able to signal the location of a cache, but this location is often not the nearest cache (*Figure 5D*). Intuitively, without barcodes, the network is unable to distinguish individual memories at nearby spatial locations. Without this barcode-mediated competition, it forms a single agglomerated memory that is inflexibly recalled.

If correlations in inputs cause memory interference, why not do away with them entirely? The Barcode Only model shows that this is not a good option. This model performs well on the Cache Presence task (*Figure 5A*, right, *Figure 5—figure supplement 1E-G*), correctly identifying the presence of all three caches and the absence of caches in other locations (*Figure 5C*). However, the Barcode Only model fails on the Cache Location task. The model is unable to recall place fields that are not precisely at its current location (*Figure 5B and D*). With greater seed input, the model can sometimes recall memories at remote locations, but these are selected randomly with no preference for nearby caches (*Figure 5D*, *Figure 5—figure supplement 1H*). Intuitively, the model cannot distinguish nearby and distant caches because it lacks input correlations, which establish the measure of proximity.

In summary, place and barcode activity play complementary roles in our model. Barcode activity functions like an index for individual memories. This function supports discriminability of memories

even when memory contents overlap – for example, for two nearby caches with correlated place activity. Barcodes also support selective recall of individual memories in our model via competitive attractor dynamics. However, the discriminability advantage of barcodes is only useful if they can be reactivated during memory recall. Spatially correlated place inputs (and the seed input) allow efficient memory retrieval by defining a measure of proximity in memory contents and support remote memory recall. The presence of both index and content signals in our model allows all of these functions to be performed by the same network, and for the trade-offs between them to be adjusted flexibly.

## Modulating recurrent strength allows the RNN to incorporate both predictive maps and barcode memory

We have constructed a model of a simple episodic memory, taking inspiration from hippocampal data (*Chettih et al., 2024*). However, the hippocampus has also been suggested to support functions beyond episodic memory. An especially influential line of prior work proposes that the hippocampus plays a role in generating predictive maps, with evidence from both experiments (including in food-caching birds; *Muller and Kubie, 1989*; *Mehta et al., 2000*; *Payne et al., 2021*; *Applegate et al., 2023*) and theory (*Blum and Abbott, 1996*; *Stachenfeld et al., 2017*; *Whittington et al., 2020*). This raises the question – is our model consistent with predictive map theories? And if so, how might predictive maps influence episodic memory recall?

Interestingly, prior work has shown that biologically realistic RNNs can generate predictive maps with structured recurrent weights (*Fang et al., 2023*). Specifically, if recurrent weights encode the transition statistics of an animal's experience, then predictive map-like activity will arise from the RNN dynamics and can be controlled by recurrent strength (analogous to the value of $r$ in *equation 3*). This prediction via structured recurrent weights invites comparison to the use of random recurrent weights in our model to generate barcodes. Inspired by this connection, we considered whether a single model could generate both predictive and barcode activity via recurrent dynamics. We constructed a hybrid model, in which recurrent weights are a blend of random weights, as above, and structured, predictive weights as in *Fang et al., 2023* (*Figure 6—figure supplement 1B*). We hypothesize that network activity changes from a predictive to a chaotic regime (supporting barcodes) with increasing recurrent strength, that is as we adjust $r$ in the range $0 < r < 1$.

We use the same circular arena for our simulations, but with the assumption that animal behavior is biased, that is the animal only moves clockwise (*Figure 6A*). We visualize population activity at each site, with units sorted by their place field. When recurrent strength is at its lowest ($r = 0$, *Figure 6B*) network activity is place-like and reflects the current location of the animal provided in its inputs. When recurrent strength is slightly increased ($r = 0.3$, *Figure 6C*), the network exhibits predictive activity. That is, neural activity of recurrent units is shifted to reflect an expected future position, relative to the current spatial position encoded in inputs. This is consistent with observations from experimental data collected from the hippocampus of animals with biased movements in a linear track (*Wilson and McNaughton, 1993*; *Mehta et al., 1997*; *Mehta et al., 2000*; *Lisman and Redish, 2009*). Finally, when recurrent strength is maximal ($r = 1.0$, *Figure 6D*) we observe barcodes, i.e. sparse activity at random positions relative to inputs. We plot spatial tuning curves of a few example RNN units under low, intermediate, and high recurrent strengths (*Figure 6E*, *Figure 6—figure supplement 1C*). Individual units can display typical place fields (blue), skewed predictive place fields (orange), and barcode activity (purple).

These results suggest that the structured and random components of recurrent connectivity can act somewhat independently of each other, with their relative contributions determined by recurrent strength. We quantify this by measuring the magnitude of the projection of population activity onto the place code, the predictive code, and barcodes as a function of recurrent strength (*Figure 6F*). At the lowest recurrent strength ($r = 0$), the population activity is solely concentrated on the place code. As recurrent strength is increased, place coding decreases and predictive activity increases with a peak around $r = 0.4$. Beyond this, both place and predictive activity decrease as barcode activity rises to a peak when $r = 1.0$.

Finally, we explore the functional implications of including predictive maps into our memory model. We first verify that model performance in the previous three-cache tasks is not disrupted by including predictive weights in the model (*Figure 6—figure supplement 2*). We then examine memory recall of a single cache for a predictive model, assuming a clockwise behavioral bias (*Figure 6G*). The model

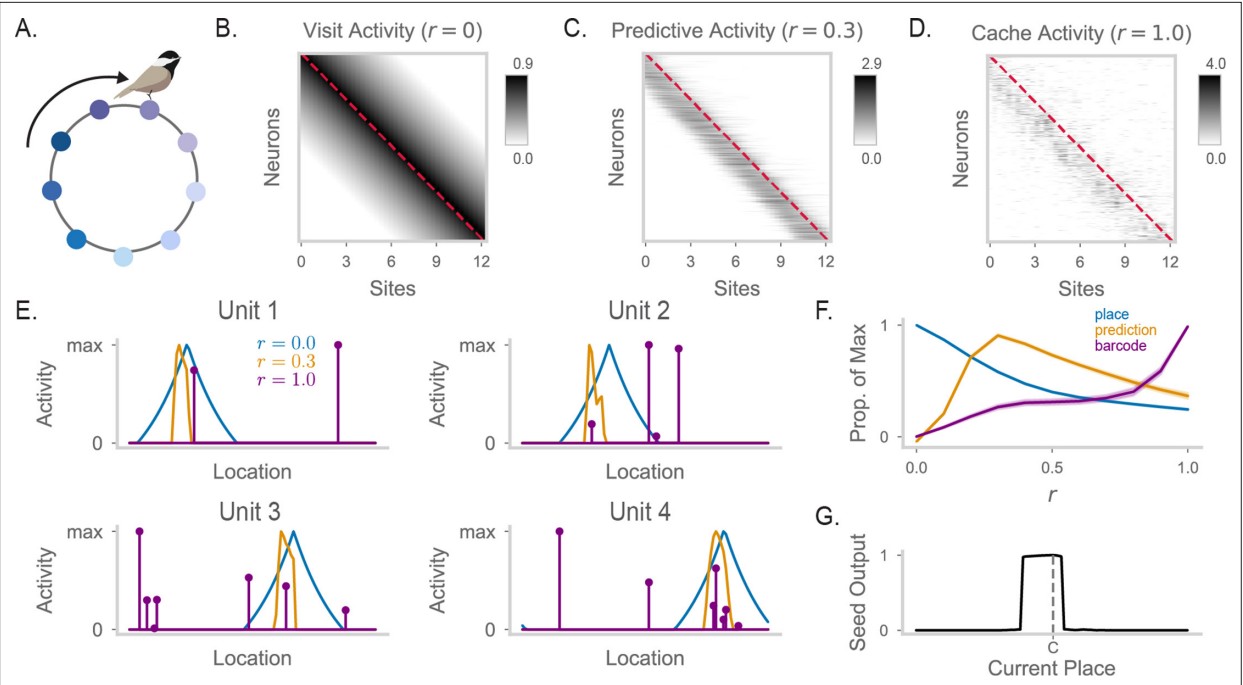

**Figure 6.** Predictive coding and barcode generation are performed by the same RNN in different dynamical regimes. (**A**) Cartoon of a chickadee in a circular track, running in a consistently clockwise direction. (**B**) Heatmap of RNN firing fields when $r = 0$, where each row corresponds to the tuning curve of one neuron across all locations. Red dashed line indicates the diagonal. (**C**) As in (**B**), but for $r = 0.3$. Here, clockwise movement corresponds to movement from site $i$ to site $i + 1$. Thus, predictive activity appears as a shift in RNN activity below the matrix diagonal. (**D**) As in (**B**), but for $r = 1.0$. Barcodes appear as random, off-diagonal structure in the activity matrix. (**E**) The firing fields of four example units across different recurrent strengths. That is, each unit's row in (**B**) is in blue, its row in (**C**) is in orange, and its row in (**D**) is in purple. Each curve is max-normalized. (**F**) Average projection strength of RNN activity onto the place code, predictive code, and barcode vectors. The place code vector is defined as activity with $r = 0$. The predictive code vector is defined as the place field of the unit at the next clockwise site, minus the projection onto the place code. The barcode vector is defined as activity with $r = 1.0$, minus the projection onto the place code. Lines show the mean over the 100 units of the mode, with shading displaying 99% confidence interval. Each line is max-normalized. (**G**) Seed output of the model with predictive weights, given the animal's location on the circular track. Here, there is only one cache, made at the halfway location of the circular track (labeled "C").

The online version of this article includes the following figure supplement(s) for figure 6:

**Figure supplement 1.** Structure of predictive weights matrix and unit activity examples.

**Figure supplement 2.** Model performance with predictive weights.

exhibits a profound skew: the cache is recalled much earlier on the path leading up to the cache location, at further distance from the cache than on the path after the cache location. This indicates that knowledge of the environment and prior experience, as reflected in predictive place activity, can shape memory recall in our model.

## Discussion

We have proposed a biologically realistic model for a simple form of episodic memory using barcodes. Our work is related to previous auto-associative memory models of the hippocampus such as Hopfield networks (***Gardner-Medwin, 1976***; ***McNaughton and Morris, 1987***; ***Marr et al., 1991***; ***Alvarez and Squire, 1994***; ***Tsodyks, 1999***), but diverges in a few critical areas. Building on ideas from hippocampal indexing theory (***Teyler and DiScenna, 1986***; ***Teyler and Rudy, 2007***), and following the discovery of barcodes (***Chettih et al., 2024***), we show how recurrent computation can implement memory indexing. Our model is further noteworthy in randomly intermixing representations of memory index and memory content in the activity of single neurons, matching experimental findings. This inter-mixing implies that single neurons cannot be definitively identified as 'place cells' or 'barcode cells', despite clear differentiation between the place code and the barcode at the population level. A further innovation of our model is the ability to control the trade-off between pattern completion and

pattern separation during memory recall, by simply turning up or down the strength of a memory content input ('search strength' in *Figure 4*). In this work, we considered only place and a single 'seed' input, but it is straightforward to generalize this to naturalistic cases where different food types are stored, or to memory contents beyond cached food. In principle, our approach would allow independent control of search strength for each potential element of memory content.

To generate barcodes during caching and retrieval without affecting place activity during visits, our model changes recurrent strength in an RNN between different behaviors. A major question is how the brain could implement such gain changes in recurrence. One possible mechanism is a change in recurrent inhibition, which is consistent with dramatic changes in the activity of inhibitory neurons observed during caching (*Chettih et al., 2024*). Neuromodulators like acetylcholine have been shown to bidirectionally modulate different inhibitory neuron subtypes (*Xiang et al., 1998*; *Lovett-Barron et al., 2014*), and proposed to control the recurrent gain of hippocampal processing (*Hasselmo, 1999*; *Hasselmo, 2006*). However, our model uses generic RNN units, and it is unclear precisely how units in the model should be mapped to real excitatory and inhibitory hippocampal neurons in the brain. Our model predicts a state change in hippocampal activity during memory formation and recall, allowing recurrent computation to generate or reactivate memory barcodes. Detailed modeling of realistic E-I networks is needed to further clarify its specific biological implementation.

Alternatively, other mechanisms may be involved in generating barcodes. We demonstrated that conventional feed-forward sparsification (*Babadi and Sompolinsky, 2014*; *Xie et al., 2023*) was highly inefficient, but more specialized computations may improve this (*Földiák, 1990*; *Olshausen and Field, 1996*; *Sacouto and Wichert, 2023*; *Muscinelli et al., 2023*). Another possibility is that barcodes are generated in a separate recurrent network upstream of the recurrent network where memories are stored. In this two-network scenario, the downstream network receives both spatial tuning and barcodes as inputs. This would not obviate the need for modulating recurrent strength in the downstream network to switch between input-driven modes and attractor dynamics. We suspect separating barcode generation and memory storage in separate networks would not fundamentally affect our conclusions.

We showed that barcodes allow for precise memory retrieval despite the presence of other correlated memories. This sharpened memory retrieval is similar to mechanisms used in key-value memory structures that are often embedded in machine learning architectures (*Graves et al., 2014*; *Graves et al., 2016*; *Sukhbaatar et al., 2015*; *Le et al., 2019*; *Banino et al., 2020*). At their simplest, these key-value memory structures consist of memory slots. Each slot consists of a memory that can be addressed via 'keys' such that their stored memory is returned as 'values'. In machine learning, key-value memory has been connected to the dot-product attention mechanism used in transformers (*Krotov and Hopfield, 2016*; *Ramsauer et al., 2020*). Interestingly, prior theoretical work has suggested neural implementations for both key-value memory and attention mechanisms, arguing for their usefulness in neural systems such as long-term memory (*Kanerva, 1988*; *Tyulmankov et al., 2021*; *Bricken and Pehlevan, 2021*; *Whittington et al., 2021*; *Kozachkov et al., 2023*; *Krotov and Hopfield, 2020*; *Gershman et al., 2025*). In this framework, the address where a memory is stored (the key) may be optimized independently of the value or content of the memory. In our model, barcodes improve memory performance by providing a content-independent scaffold that binds to memory content, preventing memories with overlapping content from blurring together. Thus, barcodes can be considered as a change in memory address, and our model suggests important connections between recurrent neural activity and key generation mechanisms. However, we note that barcodes should not be literally equated with keys in key-value systems as our model's memory is 'content-addressable'—it can be queried by place and seed inputs.

Episodic memory is often studied at a behavioral level in humans performing free or cued recall of remembered word lists (*Kahana, 2020*; *Naim et al., 2020*). Temporal context models (TCM) of episodic memory have been highly successful in accounting for the sequential order effects observed reliably in this experimental setting (*Howard and Kahana, 2002*; *Howard et al., 2005*; *Sederberg et al., 2008*), and the idea of a 'context vector' in TCM is closely related to use of barcodes as a memory index in our model. However, experiments have shown that chickadee cache retrieval does not exhibit temporal order effects (*Applegate and Aronov, 2022*), suggesting that caches at different locations are likely not linked by a temporal context as in TCM. Interestingly, caches at the same location were found to have distinct but correlated barcodes (*Chettih et al., 2024*), which could

be related to caches sharing a 'spatial context' analogous to TCM. In the present study, we did not consider memory for different caches at the same location, since it requires a mechanism for forgetting or overwriting cache memory following retrieval. Although such 'directed forgetting' is observed in chickadee behavior (*Sherry, 1984*), there is no definitive solution for Hopfield-like networks, and it is thus beyond the scope of our current work.

Our hippocampal model focused on the implementation of episodic memory. Importantly, the proposed barcode mechanism is capable of coexisting with other hippocampal functions, such as predictive coding as formalized by the successor representation (SR; *Stachenfeld et al., 2017*). Surprisingly, we found that a hybrid network can switch between SR-generating and barcode-generating modes of operation by adjusting the gain of recurrent connectivity. Further work is needed to characterize the general conditions under which barcode and SR functions do or do not mutually interfere. It is also unclear if these are separate functions of the same circuit, or if they are complementary in certain scenarios (*Schapiro et al., 2017*; *Barron et al., 2020*). For example, we found that the SR could bias barcode-mediated memory recall. In a complex environment, the Euclidean distance between two points may not correspond to its proximity in a practical sense, which the SR better captures. In this case, experience-dependent biases in memory recall can be functionally advantageous (*Dasgupta and Gershman, 2021*) and would be consistent with behavioral results (*Kahana, 1996*; *Talmi and Moscovitch, 2004*).

## Methods
### Caching task
We simulate a caching task in a circular track, similar to *Chettih et al., 2024*. The track consists of $N_s$ connected states. The goal of this task is to test how well an agent equipped with a memory model can precisely recall the locations where a cache (or memory) has been stored. Specifically, for each simulation, we first choose a set of states $C$ that will be the location where caches are made. For each state $c \in C$ we assume the agent is currently at state $c$ and allow the model to store a memory at that state. We then simulate what the output of the model is if the agent is at any of the other $N_s$ states. We continue this procedure for the remaining states in the list.

In the three-cache task, $C = \{0, m, N_s * 0.7\}$. We sweep over different values of $0 < m < N_s * 0.7$ to test the effects of site spacing. In the main figures, these three caches are made in increasing order of their location. However, we also randomly shuffle the order of caching in a supplementary figure (*Figure 4—figure supplement 1C, D*) and do not find any effects on model performance.

### Barcode model
#### Architecture
Place inputs into the model arrive from an input layer $\vec{p} \in \mathcal{R}^{N_p}$. The input layer feeds into a recurrent neural network with activity $\vec{x} \in \mathcal{R}^{N_x}$, where $N_x = N_p$. Place input units connect to recurrent units with one-to-one connections (that is, the weight matrix $J_{xi}$ from the place input layer to the recurrent network is the size $N_x$ identity matrix). Recurrent weights are encoded in the matrix $J \in \mathcal{R}^{N_x \times N_x}$. At initialization, $J \sim \mathcal{N}(\frac{\mu}{N_x}, \frac{\sigma^2}{N_x})$ where $\mu, \sigma$ are tunable hyperparameters controlling the mean and standard deviation of the distribution.

The input representing seeds arrives from a single unit $s$. The connections from $s$ to recurrent units $\vec{x}$ are encoded in the vector $\vec{j}_{sx} \in \mathcal{R}^{N_x}$. Each value of $\vec{j}_{sx}$ is sampled from the standard normal distribution.

#### Recurrent dynamics
Recurrent dynamics are run over $T$ timesteps. Let $\vec{v}_t$ be the preactivations of the recurrent population at time $t$ and $\vec{x}_t$ be the activations of the recurrent population at time $t$. That is, $\vec{x}_t = ReLU(\vec{v}_t)$. At $t = 0$, $\vec{x}_0 = \vec{v}_0 = 0$. The recurrent dynamics are defined over $\vec{v}$:

$$\frac{d\vec{v}}{dt} = \left( -\frac{\alpha}{N_x} \sum \vec{x} \right) \vec{v} + rJ\vec{x} + \vec{p} + s\vec{j}_{sx}$$

where $r \in \{0, 1\}$ is a modulatory factor that controls whether the network operates with recurrent dynamics or is purely feedforward driven. The first term in the equation corresponds to a voltage leak term where the leak at each neuron is proportional to global population activity. This effectively implements a form of divisive normalization and helps keep network activity stable even as weights are updated during the caching task. The second term in the equation represents recurrent inputs, while the last two terms represent feedforward inputs into the network.

## Input structure and timing

Place is encoded in $\vec{p}$ such that, at location $k$, each input neuron $l$ has activity $p_l = e^{\frac{-d}{\nu}}$ where $d$ is the shortest distance between $k$ and $l$ as a percentage of the circular arena size. We consider 100 evenly sampled states in this state space. Seed input $s$ may be any nonnegative scalar value.

- Place mode: Input $\vec{p}$ is active over all $T$ timesteps. Recurrence is turned off ($r = 0$), as is the seed input ($s = 0$).
- Caching mode: Input $\vec{p}$ is active over all $T$ timesteps and input $s$ is active over timesteps $[T - t_s, T]$ with strength $\lambda$. Recurrence is on ($r = 1$).
- Recall mode: Input $\vec{p}$ and input $s$ are both active over all $T$ timesteps. Recurrence is on ($r = 1$). The value of $s$ is flexibly modulated to adjust the search strength ($s \geq 0$).

## Update rule

Let $\vec{x}$ be the activations of the recurrent network at the end of recurrent dynamics (in our case, at time $T$). At each cache event, the update rule carried out is

$$\Delta J = \frac{\eta}{N_x}(\vec{x}\vec{x}^{\mathsf{T}} + \beta\vec{x}\mathbf{1}^{\mathsf{T}})$$

where $\eta$ controls the learning rate. The weight update contains a Hebbian update through $\vec{x}\vec{x}^{\mathsf{T}}$ and an inhibitory update through $\beta\vec{x}\mathbf{1}^{\mathsf{T}}$, where $\beta$ is a negative scalar. This inhibitory term causes the connections between neurons which are not co-active during caching to weaken. The update rule can also be stated in terms of the synapse $j \rightarrow i$:

$$\Delta J_{ij} \sim x_i x_j + \beta x_i$$

This is the form shown in **equation 4**.

## Network readout

To detect cache presence, we define a seed output signal. The output $y_s$ is read out from the recurrent network activity through weights $\vec{j}_{sx}$. At initialization, $\vec{j}_{sx} = 0$. Every time a cache is made, the following update rule is run: $\Delta\vec{j}_{sx} = \vec{x}$. Thus, seed output is read out as $y_s = \vec{j}_{sx}^{\mathsf{T}}\vec{x}$.

To recall cache location, we define a place field readout layer. The output $\vec{y}_p \in \mathcal{R}^{N_p}$ is read out from the recurrent network through weights $J_{yx} \in \mathcal{R}^{N_i \times N_x}$. At initialization, $J_{yx} = 0$. Every time a cache is made, the following update rule is run: $\Delta J_{yx} = \vec{p}\vec{x}^{\mathsf{T}}$. Thus, recalled cache locations are read out as $\vec{y}_p = J_{yx}\vec{x}$.

## Ablations

To construct a place code-only model, we ablate barcode generation by setting $J = 0$ at initialization. To construct a barcode-only model, we ablate the presence of place correlations in the input by setting the place input spatial scale parameter $\nu$ to a very small value $10^{-3}$ in the place encoding equation.

## Defining site distance in our simulation

To allow comparisons to data in **Chettih et al., 2024**, we first determine the number of states in our simulation that's equivalent to the distance between adjacent cache sites in **Chettih et al., 2024**. To do so, we note that the spatial correlation between population activity at two adjacent cache sites is around 0.75 (when spatial correlation profile is normalized to [0, 1]). We identify the number of states in our simulation such that the normalized correlation between visit activity is also around 0.75. We find that this is around eight states. Thus, we define eight states in our simulation as equivalent to the distance between adjacent cache sites.

## Simulating and visualizing spikes

We simulate Poisson spikes from our rate network in *Figure 2HJ*. Specifically, for a unit with rate $r$, we draw spikes from a Poisson distribution with mean and variance $r + K$. We set $K = 0.2$ to visually match observations from data.

## Spatial correlation of RNN activity for cache-retrieval pairs at different sites

To calculate correlation values as in *Figure 3D*, we simulated experiments where five sites were randomly chosen for caching and retrieval. To compare model results to the empirical data in 1E,F, which includes intrinsic neural variability, we sampled Poisson-generated spike counts from the rates output by our model. Specifically, for RNN activity $\vec{r}_i$ at location $i$, using the rates at $t = 100$ as elsewhere, we first generate a sample vector of spikes $\vec{s}_i \sim \text{Poisson}(k\vec{r}_i)$. We choose $k$ to be 0.2 to match experimental spatial correlation profiles in the 'visit-visit' condition. To get the correlation of population activity at locations $i, j$, we calculate the population Pearson correlation coefficient between $\vec{s}_i$ and $\vec{s}_j$. To generate *Figure 3D*, we repeated this simulated experiment 20 times with different random seeds and then pooled over experiments to average correlation values as a function of distance. All values are normalized by the 'visit-visit' correlation value at a site distance of 0, to match the analysis of experimental data in *Chettih et al., 2024*. This is repeated over 20 random seeds.

## Barcode model with predictive map

The barcode model with prediction differs from the default model in the initialized weight matrix. Specifically, the weight matrix $J = B + M$, where $B$ is the random Gaussian matrix of the default model. $M$ is a successor representation-like matrix (*Stachenfeld et al., 2017*) defined as

$$M = \rho(\sum_{d=0}^{D} \gamma^d T^t) + \delta$$

where $T$ is the transition probability matrix and $\gamma = 0.99$ is a temporal discount factor. We also add a scaling factor $\rho = 0.075$ and an inhibitory offset $\delta = -0.015$ is a 5000-dimensional square matrix, and we truncate the summation at $D = 300$ steps. $T$ is defined as

$$T_{ij} = \begin{cases} 1, & \text{if } j = i + 1 \\ 0, & \text{otherwise} \end{cases}$$

## Alternative model: feedforward barcode generation

We will construct a feedforward model to generate sparse, decorrelated barcodes (*Figure 2—figure supplement 2A*). Place inputs to the model are fed through a hidden expansion layer before being compressed again by an output layer to generate the barcode. We first define these layers via the random matrices $W_h \in \mathcal{R}^{M,N_p}$ and $W_o \in \mathcal{R}^{N_x,M}$ where $W_h \sim \mathcal{N}(0, \frac{1}{N_p})$ and $W_o \sim \mathcal{N}(0, \frac{1}{M})$. We make the hidden layer very large: $M = 20000$. The activity of the model in the hidden layer is described as:

$$\vec{x_h} = \text{ReLU}(W_h \vec{p} - C_\theta)$$

where $C_\theta$ is a constant chosen such that the proportion of units in $\vec{x_h}$ that are active is $\theta$. In other words, $\theta$ is a hyperparameter of the model that sets how sparse the hidden layer activity is.

The hidden layer activity is then passed through the output weights to form the barcode activity generated by the feedforward model:

$$\vec{x} = \text{ReLU}(W_o \vec{x_h} - C)$$

where $C$ is a constant chosen such that the sparsity of $\vec{x}$ matches the sparsity of barcodes generated in the default RNN barcode model.

## Alternative model: place encoding with Gaussian input weights

We simulate a version of the model with more realistic and complex place inputs. We generated the input currents to the RNN units according to a 0-mean multivariate Gaussian process. The statistics

of the Gaussian process are chosen such that the covariance of the inputs to RNN unit $i$ and unit $j$ is an exponentially decaying function of their minimum spatial distance $d$ around the circular arena: $\Sigma_{ij} = e^{-\frac{d}{0.4}}$. All other details of this alternative model are the same as in the default.

## Code

Code is publicly available on https://github.com/chingf/barcodes (copy archived at *Fang, 2024*).

## Parameter values

| Hyperparameter | | Symbol | Value |
|---|---|---|---|
| Place Inputs | Spatial scale parameter | $\nu$ | 0.2 |
| Dimensionality | Number of place input neurons | $N_i$ | 5000 |
| | Number of recurrent network neurons | $N_x$ | 5000 |
| | Number of output neurons | $N_y$ | 5000 |
| | Number of states | $N_s$ | 100 |
| RNN weight matrix | Scale of RNN initial weight mean | $\mu$ | –40 |
| | Scale of RNN initial weight standard deviation | $\sigma$ | 7 |
| RNN dynamics | Dynamics integration step | $\Delta t$ | 0.1 |
| | Dynamics divisive normalization strength | $\alpha$ | 20 |
| | Length of recurrent dynamics | $T$ | 100 |
| | Length of seed input in caching mode | $t_s$ | 5 |
| Learning parameters | Strength of seed input during caching | $\lambda$ | 3.0 |
| | Update learning rate | $\mu$ | 40 |
| | Update rule inhibition bias | $\beta$ | –0.35 |
| Analysis parameters | Readout threshold of seed output | $\kappa$ | 0.5 |

## Acknowledgements

We thank Kim Stachenfeld, Ashok Litwin-Kumar, and members of the Aronov, Stachenfeld, and Abbott labs for feedback on this work. This research was supported by the Gatsby Charitable Foundation and the Kavli Foundation and by NSF award DBI-1707398, NIH Director's New Innovator Award (DP2-AG071918), NIH Pathway to Independence Award (SC, [1]K99NS136846), NSF GRFP (CF), and DOE CSGF (JL, DE–SC0020347).

## Additional information

### Funding

| Funder | Grant reference number | Author |
|---|---|---|
| Gatsby Charitable Foundation | | Ching Fang<br>Jack W Lindsey<br>Larry F Abbott |
| The Kavli Foundation | | Ching Fang<br>Jack W Lindsey<br>Larry F Abbott |
| National Science Foundation | DBI-1707398 | Ching Fang<br>Jack W Lindsey<br>Larry F Abbott |

| Funder | Grant reference number | Author |
| --- | --- | --- |
| NIH Office of the Director | DP2-AG071918 | Dmitriy Aronov<br>Selmaan N Chettih |
| National Institute of Neurological Disorders and Stroke | 1K99NS136846 | Selmaan N Chettih |
| U.S. Department of Energy | DE-SC0020347 | Jack W Lindsey |

The funders had no role in study design, data collection and interpretation, or the decision to submit the work for publication.

## Author contributions

Ching Fang, Conceptualization, Investigation, Writing – original draft, Writing – review and editing; Jack W Lindsey, Conceptualization, Investigation, Writing – original draft; Larry F Abbott, Dmitriy Aronov, Conceptualization, Supervision, Funding acquisition, Writing – original draft; Selmaan N Chettih, Conceptualization, Supervision, Funding acquisition, Investigation, Writing – original draft, Project administration, Writing – review and editing

## Author ORCIDs

Ching Fang ⬮ https://orcid.org/0000-0003-3653-0057
Jack W Lindsey ⬮ https://orcid.org/0000-0003-0930-7327
Dmitriy Aronov ⬮ https://orcid.org/0000-0002-3717-2477
Selmaan N Chettih ⬮ https://orcid.org/0000-0003-2045-3747

Reviewer #1 (Public review): https://doi.org/10.7554/eLife.103512.3.sa1
Reviewer #2 (Public review): https://doi.org/10.7554/eLife.103512.3.sa2
Author response https://doi.org/10.7554/eLife.103512.3.sa3

# Additional files

## Supplementary files
MDAR checklist

## Data availability

The current manuscript is a computational study, so no data have been generated for this manuscript. Modeling code is publicly available on https://github.com/chingf/barcodes (copy archived at *Fang, 2024*).

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
