## [Editor Report · eLife Assessment]

This **fundamental** work substantially advances our understanding of episodic memory by proposing a biologically plausible mechanism through which hippocampal barcode activity enables efficient memory binding and flexible recall. The evidence supporting the conclusions is **convincing**, with rigorously validated computational models and alignment with experimental findings. The work will be of broad interest to neuroscientists and computational modelers studying memory and hippocampal function.

---

## [Referee Report · Reviewer #1 (Public review)]

Summary:

In this paper, the authors develop a biologically plausible recurrent neural network model to explain how the hippocampus generates and uses barcode-like activity to support episodic memory. They address key questions raised by recent experimental findings: how barcodes are generated, how they interact with memory content (such as place and seed-related activity), and how the hippocampus balances memory specificity with flexible recall. The authors demonstrate that chaotic dynamics in a recurrent neural network can produce barcodes that reduce memory interference, complement place tuning, and enable context-dependent memory retrieval, while aligning their model with observed hippocampal activity during caching and retrieval in chickadees.

Strengths:

(1) The manuscript is well-written and structured.

(2) The paper provides a detailed and biologically plausible mechanism for generating and utilizing barcode activity through chaotic dynamics in a recurrent neural network. This mechanism effectively explains how barcodes reduce memory interference, complement place tuning, and enable flexible, context-dependent recall.

(3) The authors successfully reproduce key experimental findings on hippocampal barcode activity from chickadee studies, including the distinct correlations observed during caching, retrieval, and visits.

(4) Overall, the study addresses a somewhat puzzling question about how memory indices and content signals coexist and interact in the same hippocampal population. By proposing a unified model, it provides significant conceptual clarity.

Weaknesses:

The recurrent neural network model incorporates assumptions and mechanisms, such as the modulation of recurrent input strength, whose biological underpinnings remain unclear. The authors acknowledge some of these limitations thoughtfully, offering plausible mechanisms and discussing their implications in depth. It may be worth exploring the robustness of the results to certain modeling assumptions. For instance, the choice to run the network for a fixed amount of time and then use the activity at the end for plasticity could be relaxed.

---

## [Referee Report · Reviewer #2 (Public review)]

Summary:

Striking experimental results by Chettih et al 2024 have identified high-dimensional, sparse patterns of activity in the chickadee hippocampus when birds store or retrieve food at a given site. These barcode-like patterns were interpreted as "indexes" allowing the birds to retrieve from memory the locations of stored food.

The present manuscript proposes a recurrent network model that generates such barcode activity and uses it to form attractor-like memories that bind information about location and food. The manuscript then examines the computational role of barcode activity in the model by simulating two behavioral tasks, and by comparing the model with an alternate model in which barcode activity is ablated.

Strengths of the study:

proposes a potential neural implementation for the indexing theory of episodic memory\

Provides a mechanistic model of striking experimental findings: barcode-like, sparse patterns of activity when birds store a grain at a specific location

A particularly interesting aspect of the model is that it proposes a mechanism for binding discrete events to a continuous spatial map, and demonstrates the computational advantages of this mechanism

Weaknesses:

The importance of different modeling ingredients and dynamical mechanisms could be made more clear.

---

## [Author Response]

The following is the authors’ response to the original reviews.

**Reviewer #1 (Public review):**
Summary:In this paper, the authors develop a biologically plausible recurrent neural network model to explain how the hippocampus generates and uses barcode-like activity to support episodic memory. They address key questions raised by recent experimental findings: how barcodes are generated, how they interact with memory content (such as place and seed-related activity), and how the hippocampus balances memory specificity with flexible recall. The authors demonstrate that chaotic dynamics in a recurrent neural network can produce barcodes that reduce memory interference, complement place tuning, and enable context-dependent memory retrieval, while aligning their model with observed hippocampal activity during caching and retrieval in chickadees.Strengths:(1) The manuscript is well-written and structured.(2) The paper provides a detailed and biologically plausible mechanism for generating and utilizing barcode activity through chaotic dynamics in a recurrent neural network. This mechanism effectively explains how barcodes reduce memory interference, complement place tuning, and enable flexible, context-dependent recall.(3) The authors successfully reproduce key experimental findings on hippocampal barcode activity from chickadee studies, including the distinct correlations observed during caching, retrieval, and visits.(4) Overall, the study addresses a somewhat puzzling question about how memory indices and content signals coexist and interact in the same hippocampal population. By proposing a unified model, it provides significant conceptual clarity.Weaknesses:The recurrent neural network model incorporates assumptions and mechanisms, such as the modulation of recurrent input strength, whose biological underpinnings remain unclear. The authors acknowledge some of these limitations thoughtfully, offering plausible mechanisms and discussing their implications in depth.One thread of questions that authors may want to further explore is related to the chaotic nature of activity that generates barcodes when recurrence is strong. Chaos inherently implies sensitivity to initial conditions and noise, which raises questions about its reliability as a mechanism for producing robust and repeatable barcode signals. How sensitive are the results to noise in both the dynamics and the input signals? Does this sensitivity affect the stability of the generated barcodes and place fields, potentially disrupting their functional roles? Moreover, does the implemented plasticity mitigate some of this chaos, or might it amplify it under certain conditions? Clarifying these aspects could strengthen the argument for the robustness of the proposed mechanism.

In our model, chaos is used to produce a random barcode when forming memories, but memory retrieval depends on attractor dynamics. Specifically, the plasticity update at the end of the cache creates an attractor state, and then afterwards for successful memory retrieval the network activity must settle into this attractor rather than remaining chaotic. This attractor state is a conjunction of memory content (place and seed activity) and memory index (barcode activity). Thus a barcode is ‘reactivated’ when network dynamics during retrieval settle into this cache attractor, or in other words chaotic dynamics do not need to generate the same barcode twice.

The reviewer raises an important point, which is how sensitivity to initial conditions and noise would affect the reliability of our proposed mechanism. The key question here is how noise will affect the network’s dynamics during retrieval. Would adding noise to the dynamics make memory retrieval more difficult? We thank the reviewer for suggesting we investigate this further, and below describe our experiments and changes to the manuscript to better address this topic.

We first experimented with adding independent gaussian distributed noise into each unit, drawn independently at each timestep. We analyzed recall accuracy using the same task and methods as Fig. 4F while varying the magnitude of noise. Memory recall was quite robust to this form of noise, even as the magnitude of noise approached half of the signal amplitude. This first experiment added noise into the temporal dynamics of the network. We subsequently examined adding static noise into the network inputs, which can also be thought of as introducing noise into initial conditions. Specifically, we added independent gaussian distributed noise into each unit, with the random value held constant for the extent of temporal dynamics. This perturbation decreased the likelihood of memory recall in a graded manner with noise magnitude, without dramatically changing the spatial profile. Examination of dynamics on individual trials revealed that the network failed to converge onto a cache attractor on some random fraction of trials, with other trials appearing nearly identical to noiseless results. We now include these results in the text and as a new supplementary figure, Figure S4AB.

To clarify the network dynamics and the purpose of chaos in our model, we make the following modifications in text:

Section 2.3, paragraph 2 (starting at “To store memories…”):

“…place inputs arrive into the RNN, recurrent dynamics generate an essentially random barcode, seed inputs are activated, and then Hebbian learning binds a particular pattern of barcode activity to place- and seed-related activity.”

Section 2.3, paragraph 3 (starting at “Memory recall in our network…”): As an example, consider a scenario in which an animal has already formed a memory at some location *l*, resulting in the storage of an attractor \begin{document}$\vec{a}$\end{document} into the RNN. The attractor \begin{document}$\vec{a}$\end{document} can be thought of as a linear combination of place input-driven activity \begin{document}$p(1)$\end{document}, seed input-driven activity \begin{document}$s$\end{document}, and a recurrent-driven barcode component \begin{document}$b$\end{document}. Later, the animal returns to the same location and attempts recall (i.e. sets *r* = 1, Figure 3B). Place inputs for location *l* drive RNN activity towards \begin{document}$p(1)$\end{document}, which is partially correlated with attractor \begin{document}$\vec{a}$\end{document}, and the recurrent dynamics cause network activity to converge onto attractor \begin{document}$\vec{a}$\end{document}. In this way, barcode activity \begin{document}$b$\end{document} is reactivated, along with the place and seed components stored in the attractor state, \begin{document}$p(1)$\end{document} and \begin{document}$s$\end{document}. The seed input can also affect recall, as discussed in the following section.

Section 2.4, final paragraph (starting “We further examined how model hyperparameters affected performance on these tasks”), added the following describing new results on adding noise: We found that adding noise to the network's temporal dynamics had little effect on memory recall performance (Figure S4A). However, large static noise vectors added to the network's input and initial state decreased the overall probability of memory recall, but not its spatial profile (Figure S4B).

It may also be worth exploring the robustness of the results to certain modeling assumptions. For instance, the choice to run the network for a fixed amount of time and then use the activity at the end for plasticity could be relaxed.

As described above, chaotic dynamics are necessary to generate a barcode during a cache, but not to reactivate that barcode during retrieval. During a successful memory retrieval, network activity settles into an attractor state and thus does not depend on the duration of simulated dynamics. The choice of duration to run dynamics during caching *is* important, but only insofar as activity significantly decorrelates from the initial state. We show in Figure S1B that decorrelation saturates ~t=25, and thus any random time point t > 25 would be similarly effective. We used a fixed duration runtime for caches only to avoid introducing unnecessary complication into our model.

**Reviewer #2 (Public review):**
Summary:Striking experimental results by Chettih et al 2024 have identified high-dimensional, sparse patterns of activity in the chickadee hippocampus when birds store or retrieve food at a given site. These barcode-like patterns were interpreted as "indexes" allowing the birds to retrieve from memory the locations of stored food.The present manuscript proposes a recurrent network model that generates such barcode activity and uses it to form attractor-like memories that bind information about location and food. The manuscript then examines the computational role of barcode activity in the model by simulating two behavioral tasks, and by comparing the model with an alternate model in which barcode activity is ablated.Strengths of the study:Proposes a potential neural implementation for the indexing theory of episodic memory - Provides a mechanistic model of striking experimental findings: barcode-like, sparse patterns of activity when birds store a grain at a specific locationA particularly interesting aspect of the model is that it proposes a mechanism for binding discrete events to a continuous spatial map, and demonstrates the computational advantages of this mechanism.Weaknesses:The relation between the model and experimentally recorded activity needs some clarificationThe relation with indexing theory could be made more clearThe importance of different modeling ingredients and dynamical mechanisms could be made more clearThe paper would be strengthened by focusing on the most essential aspectsComments:The model distinguishes between "barcode activity" and "attractors". Which of the two corresponds to experimentally-recorded barcodes? I would presume the attractors. A potential issue is that the attractors are, as explained in the text (l.137), conjunctions of place activity, barcode activity and "seed" inputs. The fact that the seed activity is shared across attractors seems to imply that they have a non-zero correlation independent of distance. Is that the case in the model? If I understand correctly, Fig 3D shows correlations between an attractor and barcodes at different locations, but correlations between attractors at different locations are not shown. Fig 1 F instead shows that correlations between recorded retrieval activities decay to zero with distance.More generally, the fact that the expression "barcode" is apparently used with different meanings in the model and in the experiments is potentially confusing (in the model they correspond to activity generating during caching, and this activity is distinct from the memories; my understanding is that in the experiments barcodes correspond to both caching and retrieval, but perhaps I am mistaken?).

Our intent is to use the expression “barcode” as similarly as possible between model and experimental work. The reviewer points out that the connection between barcodes in experimental and modeling work is unclear, as well as the relation of “attractors” in our model to previous experimental results. The meaning of ‘barcode’ is absolutely critical—we clarify below our intended meaning, and then describe changes to the manuscript to highlight this.

In experiments, we observed that activity during caching looked different than ordinary hippocampal activity (i.e. typical “place activity” observed during visits). Empirically there were two major differences. First, there was a pattern of neural activity which was present during every cache . This pattern was also present when birds visually inspected sites containing a cached seed, but not when visually inspecting an empty site. This is what we refer to as “seed activity”. Second, there was a pattern of neural activity which was unique to each cache. This pattern re-occurred during retrieval, and was orthogonal to place activity (see Fig. 1E-F). This is what we refer to as “barcode activity”. In summary, activity during a cache (or retrieval) contains a combination of three components: place activity, seed activity, and barcode activity.

These experimental findings are recapitulated in our model, as activity during a cache contains a combination of three components: place activity driven by place inputs, seed activity driven by seed inputs, and barcode activity generated by recurrent dynamics. Cache activity in the model corresponds to cache activity in experiments, and barcodes in the model correspond to barcodes in experiments. Our model additionally has “attractors”, meaning that network connectivity changes so that the activity generated during a simulated cache becomes an attractor state of network dynamics. “Attractors” refers to a feature of network dynamics, not a distinct activity state, and we do not yet know if these attractors exist in experimental data.

Figure 3D, as described in the figure legend, is a correlation of activity during cache and retrieval (in purple), for cache-retrieval pairs at the same or at different sites. We believe this is what the reviewer asks to see: the correlation between attractor states for different cache locations. The reviewer makes an important point: seed activity is shared across all attractors, so then why are correlations not high for all locations? This is because attractors also have a place component, which is anti-correlated for distant locations. This is evident in Fig. 3D by noticing that visit-visit correlations (black line, corresponding to place activity only) are negative for distant locations, and the correlation between attractors (purple line, cache-retrieval pairs) is subtly shifted up relative to the black line (place code only) for these distant locations. The size of this shift is due to the relative magnitude of place and seed inputs. For example, if we increase the strength of the seed input during caching (blue line), we can further increase the correlation between attractors even for quite distant sites:

To clarify the manuscript, we made the following modifications:

Section 2.2, first paragraph: We model the hippocampus as a recurrent neural network (RNN) (Alvarez and Squire, 1994; Tsodyks, 1999; Hopfield, 1982) and propose that recurrent dynamics can generate barcodes from place inputs. As in experiments, the model’s population activity during a cache should exhibit both place and barcode activity components.

Section 2.3, paragraph 3 (starting at “Memory recall in our network…”): As an example, consider a scenario in which an animal has already formed a memory at some location *l* , resulting in the storage of an attractor \begin{document}$\vec{a} $\end{document} into the RNN . The attractor \begin{document}$\vec{a} $\end{document} can be thought of as a linear combination of place input-driven activity \begin{document}$p(1)$\end{document}, seed input-driven activity \begin{document}$s$\end{document}, and a recurrent-driven barcode component \begin{document}$b$\end{document}. Later, the animal returns to the same location and attempts recall (i.e. sets *r* = 1, Figure 3B). Place inputs for *l* drive RNN activity towards \begin{document}$p(1)$\end{document}, which is partially correlated with attractor \begin{document}$\vec{a} $\end{document}, and the recurrent dynamics cause network activity to converge onto attractor \begin{document}$\vec{a} $\end{document}. In this way, barcode activity \begin{document}$b$\end{document} is reactivated as part of attractor \begin{document}$\vec{a} $\end{document}, along with the place and seed components stored in the attractor state, \begin{document}$p(1)$\end{document} and \begin{document}$s$\end{document}. The seed input can also affect recall, as discussed in the following section.

The insights obtained from the network model for the computational role of barcode activity could be explained more clearly. The introduction starts by laying out the indexing theory, which proposes that the hippocampus links an index with each memory so that the memory is reactivated when the index is presented. The experimental paper suggests that the barcode activations play the role of indexes. Yet, in the model reactivations of memories are driven not by presenting bar-code activity, but by presenting place activity (Cache Presence task) or seed activity (Cache Location task). So it seems that either place activity and seed activity play the role of indexes. Section 2.5 nicely shows that ultimately the role of barcode activity is to decorrelate attractors, which seems different from playing the role of indexes. I feel it would be useful that the Discussion reassess more critically the relationship between barcodes, indexing theory, and key-value architectures.

The reviewer highlights a failure on our part to clearly identify the connection between our findings on barcodes, indexing theory, and key-value architectures. This is another major component of the paper, and below we propose changes to the manuscript to clarify these concepts and their relationships. First, we will summarize the key points that were unclear in our original manuscript.

The reviewer equates the concept of an ‘index’ with that of a ‘query’: the signal that drives memory reactivation. This may be intuitive, but it is not how a memory index was defined in indexing theory (e.g. Teyler & DiScenna 1986). In indexing theory, the index is a pattern of hippocampal activity that is (a) generated during memory formation, (b) separate from the activity encoding memory content, and (c) linked to memory content via associative plasticity. After memory formation, a memory might be queried by activating a partial set of the memory contents, which would then drive reactivation of the hippocampal index, leading to pattern completion of memory contents. See, for example, figure 1 of Teyler and DiScenna 1986. The ‘index’ is thus not the same as the ‘query’ that drives recall.

We propose in this work that barcode activity is such an index. Indexing theory originally posited that memory content was encoded by neocortex, and memory index was encoded by hippocampus. However the experiments of Chettih et al. 2024 revealed that the hippocampus contained both memory content and memory index signals, and furthermore there was no division of cells into ‘content’ and ‘index’ subtypes. Thus our model drops the assumption of earlier work that index and content signals correspond to different neurons in different brain areas—a significant advance of our work. Otherwise, the experimentally observed barcodes and the barcodes generated by our computational model play the role of indices as originally defined.

Our original manuscript was unclear on the relationship of indexing theory and key-value systems. Our work connects diverse areas of memory models, including attractor dynamics, key-value memory systems, and memory indexing. A full account of these literatures and their relationships may be beyond the scope of this manuscript, and we note that a recent review article (Gershman, Fiete, and Irie, 2025) further clarifies the relationship between key-value memory, indexing theory, and the hippocampus. We will cite this work in our discussion as a source for the interested reader.

Briefly, a key-value memory system distinguishes between the address where a memory is stored, the ‘key’, and the content of that memory, the ‘value’. An advantage of such systems is that keys can be optimized for purposes independent of the value of each memory. The use of barcodes in our model to decorrelate memories is related to this optimization of keys in key-value memory systems. By generating barcodes and adding this to the attractor state corresponding to a cache memory, the ‘address’ of the memory in population activity is differentiated from other memories. Our work is thus consistent with the idea that hippocampus generates keys and implements a key storage system. However it is not so straightforward to *equate* barcodes with keys, as they are defined in key-value memory. As the reviewer points out, memory recall can be driven by location and seed inputs, i.e. it is content-addressable. We think of the barcode as modifying the memory address to better separate similar memories, without changing memory content, and the resulting memory can be recalled by querying with either content or barcode. Given the complex and speculative nature of these relationships, we prefer to note the salient connection of our work with ongoing efforts applying the key-value framework to biological memory, and leave the precise details of this connection to future work.

We make the following changes in the manuscript to clarify these ideas:

Introduction, first paragraph: In this scheme, during memory formation the hippocampus generates an index of population activity, and the neurons representing this index are linked with the neurons representing memory content by associative plasticity . Later, re-experience of partial memory contents may reactivate the index, and reactivation of the index drives complete recall of the memory contents.

Discussion, 4th paragraph on key-value: Interestingly, prior theoretical work has suggested neural implementations for both key-value memory and attention mechanisms, arguing for their usefulness in neural systems such as long term memory (Kanerva, 1988; Tyulmankov et al., 2021; Bricken and Pehlevan, 2021; Whittington et al., 2021; Kozachkov et al., 2023; Krotov and Hopfield, 2020; Gershman 2025). In this framework, the address where a memory is stored (the key) may be optimized independently of the value or content of the memory. In our model, barcodes improve memory performance by providing a content-independent scaffold that binds to memory content, preventing memories with overlapping content from blurring together. Thus barcodes can be considered as a change in memory address, and our model suggests important connections between recurrent neural activity and key generation mechanisms. However we note that barcodes should not be literally equated with keys in key-value systems as our model’s memory is ‘content-addresable’—it can be queried by place and seed inputs.

The model includes a number of non-standard ingredients. It would be useful to explain which of these ingredients and which of the described mechanisms are essential for the studied phenomenon. In particular:- the dynamics in Eq.2 include a shunting inhibition term. Is it essential and why?

The shunting inhibition is important as it acts to normalize the network activity to prevent runaway excitation. We hope to clarify this further by amending the following sentence in section 2.2: *“g* (·) is a leak rate that depends on the average activity of the full network, representing a form of global shunting inhibition that normalizes network activity to prevent runaway excitation from recurrent dynamics.”

- same question for the global inhibition included in the random connectivity;

The distribution from which connectivity strengths are drawn has a negative mean (global inhibition). This causes activity during caching (i.e. r = 1) to be sparser than activity during visits (i.e. r = 0), and was chosen to match experimental findings. In figures 2B and S2B we show that our model can transition between a mode with place code only, barcode only, or a mode containing both, by changing the variance of the weight distribution while holding the mean constant. We suggest clarifying this by editing the following in section 2.2, paragraph 2: “We initialize the recurrent weights from a random Gaussian distribution, \begin{document}$J \sim {N}\left(\frac{\eta}{N_X}, \frac{\sigma^{2}}{N_X}\right) $\end{document}. where 𝑁_𝑋_ is the number of RNN neurons and μ < 0, reflecting global subtractive inhibition that encourages sparse network activity to match experimental findings (Chettih et al. 2024).”

- the model is fully rate-based, but for certain figures, spikes are randomly generated. This seems superfluous.

Spikes are simulated for one analysis and one visualization, where it is important to consider noise or variability in neural responses across trials. First, for Fig. 2H,J, we generated spikes to allow a visual comparison to figures that can be easily generated from experimental data. Second, and more significantly, for the analysis underlying Fig. 3D, it is essential to simulate variability in neural responses. Because our rate-based models are noiseless, the RNN’s rate vector at site distance = 0 will always be the same and result in a correlation of 1 for both visit-visit and cache-retrieval. However, we show that, if one interprets the rate as a noisy Poisson spiking process, the correlation at site distance = 0 between a cache-retrieval pair is higher than that of two visits. This is because under a Poisson spiking model, the signal-to-noise ratio is higher for cache-retrieval activity, where rates are higher in magnitude. The greater correlation for a cache-retrieval pair at the same site, relative to visits at the same site, is an experimental finding that was critical for our model to reproduce. We detail clarifications to the manuscript below in response to the reviewer’s following and related question.

How are the correlations determined in the model (e.g., Fig 2 B)? The methods explain that they are computed from Poisson-generated spikes, but over which time period? Presumably during steady-state responses, but are these responses time-averaged?

The reviewer points out a lack of clarity in our original manuscript. Correlations for events (caches, retrievals and visits) at different sites are calculated in two sections of the paper (2B, 3D), for different purposes and with slight differences in methods:

- For figure 2B, no spikes are simulated. Note that the methods mentioning poisson spike generation specify only Fig. 2H,J and Fig. 3D. We simply take the network’s rate vector at timestep t=100 (when the decorrelating effect of chaotic dynamics has saturated, S1A-B) and correlate this vector when generated at different locations. We now clarify this in the legend for Figure 2B: “We show correlation of place inputs (gray) and correlation of the RNN's rate vector at t = 100 (black).”

- For Figure 3D, we want to compare the model to empirical results from Chettih et al. 2024, and reproduced in this paper in Fig. 1E-F. These empirical results are derived from correlating vectors of spiking activity on pairs of single trials, and are thus affected by noise or variability in neural responses as described in our response to the reviewer’s previous question. We thus took the RNN’s rate vector at t=100 and simulated spiking data by drawing samples from a poisson distribution to get spike counts. Our original manuscript was unclear about this, and we suggest the following changes:

- Legend for Figure 3D: D. Correlation of Poisson-generated spikes simulated from RNN rate vectors at two sites, plotted as a function of the distance between the two sites.

- Section 2.3, last paragraph: Population activity during retrieval closely matches activity during caching, and is substantially decorrelated from activity during visits (Figure 3C). To compare our model with the empirical results reproduced in Figure 1E,F, we ran *in silico* experiments with caches and retrievals at varying sites in the circular arena. We simulated Poisson-generated spikes drawn from our network's underlying rates to match the intrinsic variability in empirical data (see Methods).

- Methods, subsection Spatial correlation of RNN activity for cache-retrieval pairs at different sites: To calculate correlation values as in Figure3D, we simulated experiments where 5 sites were randomly chosen for caching and retrieval. To compare model results to the empirical data in Fig. 1E,F, which includes intrinsic neural variability, we sampled Poisson-generated spike counts from the rates output by our model. Specifically, for RNN activity \begin{document}$\vec{r_i}$\end{document} at location i, using the rates at t=100 as elsewhere, we first generate a sample vector of spikes…

I was confused by early and late responses in Fig 2 C. The text says that the activity is initialized at zero, so the response at t=0 should be flat (and zero). More generally, I am not sure I understand why the dynamics matter for the phenomenon at all, presumably the decorrelation shown in Fig 2B depends only on steady state activity (cf previous question).

Thanks for catching this mistake. The legend has been updated to indicate that the ‘early’ response is actually at t=1, when network activity reflects place inputs without the effects of dynamics. The reviewer is correct that we are primarily interested in the ‘late’ response of the network. All other results in the paper use this late response at t=100. As shown in Fig. S2A,B, this timepoint is not truly a steady state, as activity in the network continues to change, but the decorrelation of network activity with place-driven activity has saturated.

We include the early response in Fig. 2C for visual comparison of the purely place-driven early activity with the eventual network response. It is also relevant since, as the reviewer points out above, there is a shunting inhibition term in the dynamics that is present during both low and high recurrent strength simulations.

Related to the previous point, the discussion of decorrelation (l.79 - 97) is somewhat confusing. That paragraph focuses on chaotic activity, but chaos decorrelates responses across different time points. Here the main phenomenon is the decorrelation of responses across different spatial inputs (Fig 2B). This decorrelation is presumably due to the fact that different inputs lead to different non-trivial steady-state responses, but this requires some clarification. If that is correct, the temporal chaos adds fluctuations around these non-trivial steady-state responses, but that alone would not lead to the decorrelation shown in Fig 2B.

We agree with the reviewer that chaotic activity produces a decorrelation across time points. Because of chaotic dynamics, network activity does not settle into a trivial steady-state, and instead evolves from the initial state in an unpredictable way. The network does not settle into a steady-state pattern, but both the decorrelation of network state with initial state and the rate of change in the network state saturate after ~t=25 timesteps, as shown in Fig. S2A-B.

The initial activity for nearby states is similar, due to them receiving similar place inputs.

Because network activity is chaotically decorrelated from this initial state by temporal dynamics, ‘late stage’ network activity between nearby spatial states is less correlated than ‘early stage’ activity. Thus the temporal decorrelation produces a spatial decorrelation. We believe that the changes we have introduced to the manuscript in revision will make this point clearer in our resubmission.

A key ingredient of the model is that the recurrent interactions are switched on and off between "caching" and "visits". The discussion argues that a possible mechanism for this is recurrent inhibition (l.320), which would need to be added. However two forms of inhibition are already included in the model. The text also says that it is unclear how units in the model should be mapped onto E and I neurons. However the model makes explicit assumptions about this, in particular by generating spikes from individual neurons. Altogether, I did not find that part of the Discussion convincing.

We agree with the reviewer that this section is a limitation of our current work, and in fact it is an ongoing area of future research. However we think the advances in this current work warrant publication despite this topic requiring further research. We attempted to discuss this limitation explicitly, and note that the other reviewer pointed this section out as particularly helpful. We do not think it is problematic for a realistic model of the brain to ultimately include 3, or even more forms of inhibition. We do not think that poisson-generated spikes commit us to interpreting network units as single neurons. Spikes are not a core part of our model’s mechanism, and were used only as a mechanism of introducing variability on top of deterministic rates for specific analyses. Furthermore one could still view network units as pools of both E and I spiking neurons. We would welcome further recommendations the reviewer believes are important to note in this section on our model’s limitations.

On lines 117-120 the text briefly mentions an alternate feed-forward model and promptly discards it. The discussion instead says that a "separate possibility is that barcodes are generated in a circuit upstream of where memories are stored, and supplied as inputs to the hippocampal population", and that this possibility would lead to identical conclusions. The two statements seem a bit contradictory. It seems that the alternative possibility would replace the need for switching on and off recurrent interactions, with a mechanism where barcode inputs are switched on and off. This alternate scenario is perhaps more plausible, so it would be useful to discuss it more explicitly.

We apologize for the confusion here, which seems to be due to our phrasing in the discussion section. We do reject the idea that a simple feed-forward model could generate the spatial correlation profile observed in data, as mentioned in the text and included as Fig. S2. Our statement in the discussion may have seemed contradictory because here we intended to discuss the possibility that an upstream area generates barcodes, for example by the chaotic recurrent dynamics proposed in our work, while a downstream network receives these barcodes as inputs and undergoes plasticity to store memories as attractors. We did not intend to suggest any connection to the feedforward model of barcode generation, and apologize for the confusion. Our claim that this ‘2 network’ solution would lead to similar conclusions is because the upstream network would need an efficient means of barcode generation, and the downstream network would need an efficient means of storing memory attractors, and separating these functions into different networks is not likely to affect for example the advantage of partially decorrelating memory attractors. Moreover, the downstream network would still require some form of recurrent gating, so that during visits it exhibits place activity without activating stored memory attractors!

We thus chose a 1 network instead of a 2 network solution because it was simpler and, we believe, more interesting. It is challenging in the absence of more data to say which is more plausible, thus we wanted to mention the possibility of a 2 network solution. We suggest the following changes to the manuscript:

- Discussion, 3rd paragraph: “Alternatively, other mechanisms may be involved in generating barcodes. We demonstrated that conventional feed-forward sparsification (Babadi and Sompolinsky, 2014; Xie et al., 2023) was highly inefficient, but more specialized computations may improve this (Földiak, 1990; Olshausen and Field, 1996; Sacouto and Wichert, 2023; Muscinelli et al., 2023). Another possibility is that barcodes are generated in a separate recurrent network upstream of the recurrent network where memories are stored. In this 2-network scenario, the downstream network receives both spatial tuning and barcodes as inputs. This would not obviate the need for modulating recurrent strength in the downstream network to switch between input-driven modes and attractor dynamics. We suspect separating barcode generation and memory storage in separate networks would not fundamentally affect our conclusions.”

As a minor note, the beginning of the discussion states that the presented model is similar to previous recurrent network models of the hippocampus. It would be worth noting that several of the cited works assign a very different role to recurrent interactions: they generate place cell activity, while the present model assumes it is inherited from upstream inputs.

We are not sure how best to modify the paper to address this suggestion. As far as we know, all of the cited models which deal with spatial encoding do assume that the hippocampus receives a spatially-modulated or spatially-tuned input. For example, the Tsodyks 1999 paper cited in this paragraph uses exponentially-decaying place inputs to each neuron highly similar to our model. Furthermore we explore how our model would perform if we change the format of spatial inputs in Fig. S4, and find key results are unchanged. It is unclear how hippocampal place fields could emerge without inputs that differentiate between spatial locations. We think it is appropriate to highlight the similarity of our model to well known hopfield-type recurrent models, where memories are stored as attractor states of the network dynamics.

On the other hand, we agree that a common line of hippocampal modeling proposes that recurrent interactions reshape spatial inputs to produce place fields. This often arises in the context of hippocampus generating a predictive map, where inputs may be one-hot for a single spatial state, in a grid cell-like format, or a random projection of sensory features. We attempted to address this in section 2.6, using a model which superimposes the random connectivity needed for barcode generation with the structured connectivity needed for predictive map formation. We found that such a model was able to perform both predictive and barcode functions, suggesting a path forward to connecting different lines of hippocampal modeling in future work.